# Conserve-Update-Revise to Cure Generalization and Robustness Trade-off in Adversarial Training

**Shruthi Gowda**[1,2], **Bahram Zonooz**[* 2,3] **& Elahe Arani**[* 2,4]
[1]NavInfo Europe  [2]Eindhoven University of Technology  [3]TomTom  [4]Wayve
s.gowda@tue.nl, b.zonooz@tue.nl, e.arani@gmail.com

## Abstract

Adversarial training improves the robustness of neural networks against adversarial attacks, albeit at the expense of the trade-off between standard and robust generalization. To unveil the underlying factors driving this phenomenon, we examine the layer-wise learning capabilities of neural networks during the transition from a standard to an adversarial setting. Our empirical findings demonstrate that selectively updating specific layers while preserving others can substantially enhance the network's learning capacity. We therefore propose CURE, a novel training framework that leverages a gradient prominence criterion to perform selective conservation, updating, and revision of weights. Importantly, CURE is designed to be dataset- and architecture-agnostic, ensuring its applicability across various scenarios. It effectively tackles both memorization and overfitting issues, thus enhancing the trade-off between robustness and generalization and additionally, this training approach also aids in mitigating "robust overfitting". Furthermore, our study provides valuable insights into the mechanisms of selective adversarial training and offers a promising avenue for future research. [1]

## 1 Introduction

The susceptibility of deep neural networks (DNNs) to adversarial attacks (Szegedy et al., 2014; Goodfellow et al., 2015) continues to present a substantial challenge in the field. Adversarial training has emerged as a promising strategy to enhance the robustness of DNNs against adversarial attacks (Madry et al., 2018; Zhang et al., 2019; Tramèr et al., 2018; Wang et al., 2019). However, transitioning from standard training with natural images to adversarial training introduces distinct behavior patterns. Despite the benefits of adversarial training in improving robustness, it often results in compromised performance on clean images, creating a noticeable trade-off between standard and adversarial generalization (Raghunathan et al., 2019). Another intriguing observation is that, in contrast to the standard setting, longer durations of adversarial training can paradoxically lead to reduced test performance. This generalization gap in robustness between training and testing data, commonly referred to as robust overfitting (Rice et al., 2020), is prevalent in adversarial training. Therefore, it is imperative to gain a deeper understanding of the underlying factors driving these behaviors to advance the development of reliable and trustworthy AI systems.

Few studies have attempted to understand learning behavior in an adversarial setting. Nakkiran (2019) show that the capacity of the network needs to be increased for better generalization, while others have found that this is not always the case (Huang et al., 2021). Additionally, suggestions include the need for larger amounts of data, labeled (Schmidt et al., 2018) or unlabeled (Carmon et al., 2019; Najafi et al., 2019), for better performance. Other works have revealed that the discord between natural and adversarial accuracy is caused by differently trained feature representations (Tsipras et al., 2018; Ilyas et al., 2019). However, numerous aspects of neural network learning behavior when confronted with adversarial samples remain unexplored. Exploring how networks adapt to both natural and adversarial data can yield valuable insights into their learning dynamics and capabilities.

---

[*]Contributed equally.
[1]The code is available at: https://github.com/NeurAI-Lab/CURE.

Our study empirically investigates neural network learning capabilities during the transition from standard to adversarial settings. Rather than considering the entire network as a unified entity, we seek to identify fixable weights that maintain performance on natural data and updatable weights for learning adversarial data without a significant accuracy drop in the other distribution. Layer-wise analysis of network properties regarding weight updates demonstrates that not all weights need modification for effective learning. Selectively updating specific weights while preserving information in others can achieve a better performance balance on standard and adversarial distributions.

Inspired by the findings of the empirical analysis, we propose a novel scheme for adversarial training characterized by an adaptive and dynamic parameter update mechanism. By assessing the significance of gradients, our method enables the network to retain valuable information in specific weights while updating only those necessary for the data distribution. We introduce a novel *Robust Gradient Prominence* which guides the decision-making process regarding which parameters to update and which to keep fixed. Our proposed method, *CURE*, consists of conservation (maintaining a subset of parameters for retention of natural knowledge), updation (modifying a subset of parameters to learn adversarial knowledge), and revision (consolidating knowledge periodically).

CURE achieves a better trade-off between standard and robust generalization compared to the state-of-the-art (SOTA) methods on different architectures and datasets (Figure 1). Notably, two existing research areas focused on enhancing the trade-off and mitigating robust overfitting often do not synergize. For instance, regularization-based approaches like TRADES (Zhang et al., 2019), intended to enhance the trade-off, still contend with robust overfitting issues. Conversely, adversarial weight perturbation methods (Wu et al., 2020), designed to mitigate robust overfitting, exhibit lower natural accuracy, thus indicating reduced overall generalization. In contrast, CURE not only achieves an improved trade-off but also effectively reduces overfitting, surpassing methods explicitly designed to address this issue. Additionally, models trained using CURE exhibit

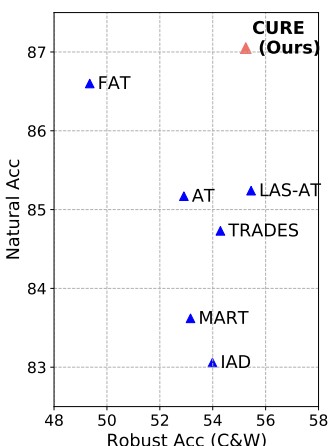

Figure 1: Generalization robustness trade-off on WideResNet-34-10 and CIFAR-10. CURE displays a better trade-off between standard and robust (C&W) performance.

higher robustness against adversarial attacks, requiring stronger and more semantically meaningful attacks to deceive the network than other methods (Figure 8). Overall, our proposed approach offers a promising direction for understanding the broader concepts of generalization and robustness in DNNs and paves the way for improved model performance in real-world scenarios.

## 2 BACKGROUND

**Standard Training:** Given a natural dataset $D$ with input and label pairs $(x, y) \in D$, standard training (ST) involves minimizing the objective function with respect to the model parameters $\theta$. Deep neural networks trained on clean or natural data exhibit high generalization on test data. However, the network generates incorrect predictions if the natural input is imperceptibly modified.

$$\text{ST:} \quad \theta^* = \min_{\theta} \mathbb{E}_{(x,y) \sim \mathcal{D}} \left[ \mathcal{L}_\theta(x, y) \right]. \tag{1}$$

Adversarial examples are perturbed inputs that result in misclassifications. The perturbations are carefully crafted imperceptible noise that, when added to the inputs, visually results in a similar image but can fool the neural network. For an input $x$ and label $y$, an adversarial sample can be generated by adding a perturbation $\delta$ such that,

$$\delta = \arg\max_{\delta \in \Delta} \mathcal{L}_\theta(x + \delta, y) \tag{2}$$

where $\Delta$ is usually a simple convex set, such as an $\ell_p$ ball.

**Adversarial Training:** AT has emerged as one of the effective techniques to make neural networks more robust and less susceptible to such attacks. Adversarial examples are generated during training to improve predictions on these images to produce a robust network. Standard Adversarial Training

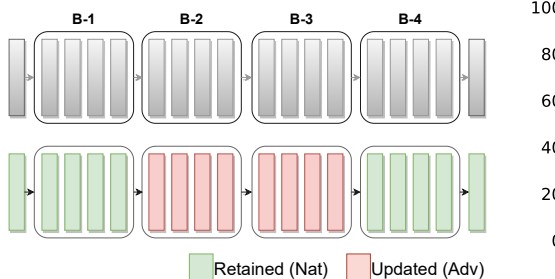 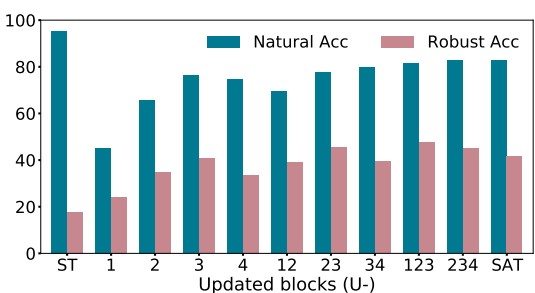

Figure 2: (a) The four blocks of ResNet-18 are considered for the layer-wise study. All layers are trained with natural images (ST) first. The second row shows the example architecture for U-23, where the first and last blocks are frozen and the second and third are updated while training adversarially. (b) Standard generalization and robustness of different blocks of ResNet-18 on CIFAR-10 dataset
.

(AT)(Madry et al., 2018) is formulated as a min-max optimization problem,

$$\text{AT:} \quad \theta^* = \min_{\theta} \mathbb{E}_{(x,y) \sim \mathcal{D}} \left[ \max_{\delta \in \Delta} \mathcal{L}_{\theta}(x + \delta, y) \right], \tag{3}$$

where the inner maximization generates strong adversarial examples for each input during training, and the outer minimization trains the model on these adversarial examples to improve robustness.

Several AT techniques have been proposed that include both natural and adversarial images during training. Adversarial regularization methods (Goodfellow et al., 2015) include regularization objectives in addition to classification loss. TRADES (Zhang et al., 2019) introduced a regularization objective between the probabilities of natural and their adversarial counterparts and MART (Wang et al., 2019) adds an emphasis on the influence of misclassified examples. Multi-model approaches utilize more than one network or ensemble-based training techniques to improve the trade-off. Many semi-/unsupervised learning-based AT techniques (Schmidt et al., 2018) have emerged. These techniques leverage additional data, increased network capacity, or network ensemble strategies and do not examine the learning patterns to perform adaptive training.

**Robust Overfitting:** In contrast to ST, where prolonged training leads to near-zero training errors and decreased test loss, adversarial training exhibits a distinct behavior characterized by an increase in test errors beyond a specific threshold, referred to as "robust overfitting" (Rice et al., 2020). This challenging issue has proven resistant to various conventional mitigation methods, including regularization and data augmentation. Notably, recent advances have introduced several strategies for alleviating robust overfitting in DNNs, including early stopping Rice et al. (2020), semi-supervised learning (Carmon et al., 2019), as well as data interpolation techniques (Lee et al., 2020; Chen et al., 2021). Furthermore, alternative approaches leverage sample re-weighting (Zhang et al., 2021), weight perturbation (Wu et al., 2020) and weight smoothing techniques (Chen et al., 2020; Hwang et al., 2021) to combat robust overfitting.

## 3 LAYER-WISE CONSERVATION AND UPDATION: EMPIRICAL ANALYSIS

Despite extensive research efforts addressing adversarial training, a persistent knowledge gap hampers our comprehension of the network's learning behavior during the transition from standard to adversarial training. This limited understanding hinders the efficacy of proposed solutions, often neglecting the unique learning capacities of network layers. Understanding how weight initialization and updates affect the loss landscape and convergence is critical, particularly as the network adapts to diverse data distributions. A layer-wise network analysis can help identify which weights are suited for retaining information and which possess greater learning capacity. Re-initialization has been explored in ST (Li et al., 2020; Alabdulmohsin et al., 2021), but its effect on AT has not been thoroughly investigated. We perform selective freezing, re-initialization, and update of specific weights to gain deeper insight into the network's behavior. By examining these factors, we can uncover valuable insights that further enhance the effectiveness of training strategies.

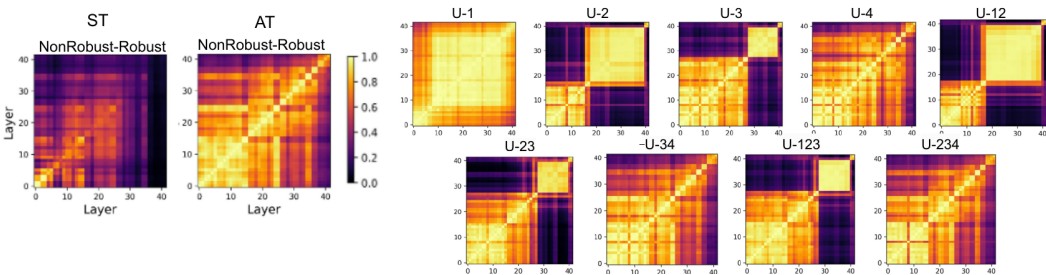

Figure 3: Representation similarity between robust and nob-robust features in ResNet-18 trained on the CIFAR-10 dataset.

**Setup**: We investigate the layer-wise properties of a network by first training it in the standard way on natural images and then transitioning to the adversarial setting. We reinitialize different layers in each experiment while keeping the rest of the network fixed. We utilize ResNet-18 as the network, with four blocks and an additional linear classifier layer. The notation U-b represents the update of block "b" while keeping the rest of the network frozen. We create 12 different combinations (U-1, U-2, U-3, U-4, U-12, U-23, U-34, U-123, U-234). The example of U-23 is shown in Figure2(a). More details of the empirical study are provided in Appendix, Section G.

## 3.1 ADVERSARIAL ROBUSTNESS

The natural and adversarial accuracy of multiple models on the CIFAR-10 dataset is shown in Figure 2(b). The 'ST' and 'SAT' represent the accuracy of standard training and adversarial training (Madry et al., 2018) respectively. Our study on the learning behavior of neural networks during the transition from ST to AT uncovers intriguing properties of the layers. The results show that training only the early layers of a network adversarially can cause reduced performance on both data distributions. As the network acquires knowledge on a new distribution, the existing information undergoes a certain degree of compromise, characterized as overwriting. Notably, we observe that updating specific layers exert a more pronounced impact on accuracy. Updating the early layers (U-1) results in a decrease in accuracy, indicating that enhancing the trade-off is not facilitated by updating these layers. In contrast, making later layers trainable leads to a milder decline in accuracy. This is attributed to the fixed representations learned in the earlier layers aiding the subsequent layers to adapt to the new distribution, reflecting increased learning capacity. These findings suggest that training the entire network without selectively updating the parameters may not be optimal for training.

## 3.2 REPRESENTATIONS ALIGNMENT

Visualizing the features learned on natural and adversarial data and examining the differences helps to understand the representation alignment. We employ a representation similarity metric, Centered Kernel Alignment (CKA) (Kornblith et al., 2019). Figure 3 visually depicts the similarity between natural and robust representations for each model. In ST, we observe significant dissimilarity in representations, indicating the inherent challenge for naturally trained models to perform well on adversarial data. In contrast, AT results in similar representations in the early layers but divergence in the final layer, driven by perturbations designed to alter decision boundaries. Models that selectively update primarily the final layers exhibit greater similarity and fewer block structures in their representations. Optimizing parameters to promote alignment and learn a generic representation between the distributions holds promise for achieving a better balance.

## 3.3 ROBUST OVERFITTING

Analysis of adversarial test accuracy over the course of long training reveals a consistent trend of declining accuracy beyond a certain point. Figure 4 illustrates the test accuracy curves for all models. The base (AT) model prominently exhibits overfitting. When we selectively update early layers, we observe a reduction in overfitting tendencies, albeit at the cost of lower overall accuracy (U-1, U-2, and U-12). The middle layers play an important role in balancing between reducing overfitting and

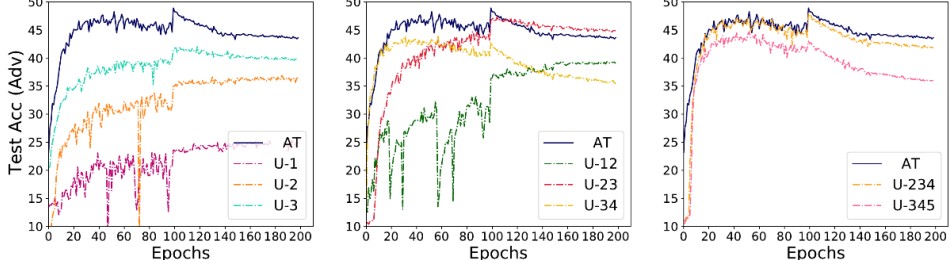

Figure 4: Adversarial test accuracy during training after layer-wise updation and conservation of ResNet-18 blocks on CIFAR-10 dataset.

having higher test performance, and the U-23 model stands out with relatively stable test accuracy. This suggests that selectively updating the layers plays a role in mitigating robust overfitting while maintaining competitive performance levels.

*The empirical findings presented in this study underscore the significance of dynamic updation in neural networks and its impact on overall performance, thus opening up new possibilities for developing more effective AT methods.*

## 4 PROPOSED METHOD

**Hypothesis**: Our empirical study on neural networks exposed to adversarial examples has led to several key insights. We postulate that the traditional approach of training the entire network as a unit and updating all weights is suboptimal when learning different data distributions. Instead, selectively updating certain weights and conserving others can lead to more effective utilization of the network's learning capabilities. By considering the retention and learning capabilities of the network, we can achieve a better balance between natural and adversarial robustness, reduce robust overfitting, and promote increased representation similarity between robust and non-robust features. Based on these findings, we propose a new AT method incorporating these considerations for enhanced performance.

**Conserve-Update-Revise**: Our method is based on three key components: Conservation (of knowledge from natural data), Updation (of knowledge from adversarial data), and REvision (of consolidated knowledge) - CURE. Instead of manually fixing and updating the layers, as in our initial empirical analysis, we devise a technique to dynamically update the weights of the network in each layer based on an importance criterion.

Consider a network $f$ and a classifier $g$ parameterized by $\theta$. $(x_{nat}, y) \in D$ represents the natural images and labels. The network is trained on natural images using the standard training schema. We use this pre-trained network and switch to the adversarial training setting. Adversarial samples $x_{adv}$ are generated during training for each clean sample $x_{nat}$ by adding a perturbation $\delta^*$. For training in the adversarial setting, we utilize a classification loss to improve the performance on natural images and an adversarial regularization loss to encourage the network to produce similar probabilities for both natural and adversarial samples. As seen in Figure 3, a better feature similarity between natural and adversarial distributions can result in robust models. To minimize the distance between the predictions of these distributions, we utilize the KL-divergence objective;

$$\delta^* = \arg\max_{\delta \in \Delta}\Big[\mathcal{D}_{KL}(p(x_{nat}; \theta)||p(x_{nat} + \delta; \theta))\Big], \tag{4}$$

$$\mathcal{L}_{adv} = \mathcal{L}_{CE}(x_{nat}; \theta) + \mathcal{D}_{KL}(p(x_{nat}; \theta)||p(x_{adv}, \theta)). \tag{5}$$

Instead of updating all the weights, we want to update a subset with the most significant impact on accuracy. Conserving some of the weights allows for more learning capacity, thus helping avoid memorization and overwriting and achieving a better trade-off. To increase performance on both distributions, we introduce a novel "Robust Gradient Prominence" objective to decide on the weights to be updated. The idea is that the feature representations that have the most impact on the model's predictions will have a greater influence on the gradients of the model parameters. Thus, the gradient

prominence score can help identify the essential weights whose activation representations can impact the predictions. When dealing with two different distributions, gradient prominence can measure which weights contribute more to the joint distribution of both natural and adversarial accuracy and which can skew the results.

*Robust Gradient Prominence* (RGP) aims to achieve this by utilizing the gradient importance of both natural and adversarial samples. RGP dynamically determines which weights to update and which ones to freeze in each layer of the network.

$$\mathcal{RGP}(w) = \alpha \left\| \frac{\partial \mathcal{L}(x_{nat}; \theta)}{\partial w} \right\| + (1 - \alpha) \left\| \frac{\partial \mathcal{L}(x_{adv}; \theta)}{\partial w} \right\|, \qquad (6)$$

where $\mathcal{L}(x_{nat}, \theta)$ and $\mathcal{L}(x_{adv}, \theta)$ are the cross-entropy losses on natural and adversarial examples, respectively. $\alpha$ helps balance the importance between natural vs. adversarial distributions.

Based on the RGP score, a gradient mask is created by removing a fraction ($p\%$) of weights with a low prominence measure. The gradient mask is applied to update only the most important weights. RGP ensures that we (1) conserve past knowledge by preventing a fraction of weights from updating, (2) learn new knowledge by updating weights that have the capacity to learn without overwriting, and (3) utilize the network's capacity efficiently to achieve balanced performance.

During training with natural and adversarial images, the model's performance may lean towards one of the distributions at different points in the training process. We introduce the revision stage to consolidate knowledge across all these patterns and obtain a more balanced performance. A revision model is updated with a stochastic momentum update (SMU) of the training model at a revision rate $r$ and a revision decay factor of $r$, $\theta_{rev} \leftarrow SMU(\theta; d)$.

$$\theta_{rev} = d \cdot \theta_{rev} + (1 - d) \cdot \theta \; ; \text{if } sample \sim U(0, 1) < r. \qquad (7)$$

Stochastic updates to the revision help accumulate knowledge throughout the training, and this consolidated information is shared with the training model. The adversarial training receives an additional boost from this knowledge revision of the accumulated knowledge. A consistency regularization objective regularizes the knowledge flow from the revision model to the training model.

$$\mathcal{L}_{CR} = \mathcal{D}_{KL}(p(x_{nat}; \theta_{rev}) || p(x_{nat}; \theta)) + \mathcal{D}_{KL}(p(x_{adv}; \theta_{rev}) || p(x_{adv}; \theta)). \qquad (8)$$

The overall training objective is $\mathcal{L} = \mathcal{L}_{adv} + \gamma \mathcal{L}_{CR}$. Algorithm is detailed in Appendix, Section A.

CURE improves adversarial training by conserving weights with lower RGP scores, updating weights that are more accountable for better performance on both natural and adversarial data, and additionally incorporating a knowledge revision from the consolidated and stable model.

## 5 RESULTS

**Experimental Setup:** We benchmark our method on various architectures, datasets, and attacks. We introduce the Natural-Robustness Ratio (NRR) metric to quantify the trade-off between generalization and robustness. All experimental details, hyperparameters, metrics, and analysis details are provided in Appendix Sections F and G.

The performance of CURE against various SOTA methods on CIFAR-10 dataset on two different architectures is tabulated in Table 1. 'Nat' denotes accuracy on the clean test set, and evaluations include PGD20, C&W, and AutoAttacks (AA). The table also features methods that incorporate more than one model in their training framework (the second part). Notably, CURE achieves the highest NRR among all methods, showcasing superior performance in balancing robustness and natural accuracy. For instance, when considering the WideResNet-34-10 model, CURE outperforms TRADES by 2.5% on PGD, FAT by 10% on AA, and multiple-teacher-based technique IAD by 2.3% on C&W attacks, all while maintaining the highest natural accuracy.

Further, we present the performance of the ResNet-18 network on both CIFAR-100 and SVHN datasets in Table 2. CURE outperforms all other methods, even in the face of challenging attacks like PGD with higher steps and C&W. When subjected to the C&W attack on SVHN, CURE exhibits a relative improvement of 10% over TRADES, 22% over MART, and 8% over ST, while maintaining the highest natural accuracy, surpassing TRADES by 3%, MART by 5% and ST by 2.3%.

Table 1: Performance comparisons on the CIFAR-10 dataset encompassing various architectures and attacks. The second part highlights methods that employ multiple networks in their approach. NRR quantifies the trade-off between natural performance and C&W attack results.

| Method | | WideResNet-34-10 | | | | | ResNet-18 | | | | |
|--------|--------|-------|-------|-------|-------|-------|-------|-------|-------|-------|-------|
| | | Nat | PGD20 | AA | C&W | NRR | Nat | PGD20 | AA | C&W | NRR |
| AT | ICLR'18 | 85.17 | 55.08 | 44.04 | 52.91 | 65.27 | 82.78 | 51.30 | 44.63 | 49.72 | 62.12 |
| TRADES | ICML'19 | 84.73 | 56.82 | 52.95 | 54.29 | 66.17 | 82.41 | 52.76 | 48.37 | 50.43 | 62.57 |
| MART | ICLR'20 | 83.62 | 56.74 | 51.23 | 53.16 | 64.99 | 80.70 | 54.02 | 47.49 | 49.35 | 61.24 |
| FAT | ICML'20 | 86.60 | 49.86 | 47.48 | 49.35 | 62.87 | 87.72 | 46.69 | 43.14 | 49.66 | 63.41 |
| ST-AT | ICLR'23 | 84.92 | 57.73 | 53.54 | - | - | 83.10 | 54.62 | 50.50 | 51.43 | 63.53 |
| ACT | BMVC'20 | 87.10 | 54.77 | - | - | - | 84.33 | 55.83 | - | - | - |
| ARD | AAAI'20 | 85.18 | 53.79 | - | - | - | 82.84 | 51.41 | - | - | - |
| IAD | ICLR'22 | 83.06 | 56.17 | 52.68 | 53.99 | 65.44 | 80.63 | 53.84 | 50.17 | 51.60 | 62.92 |
| LAS-AT | CVPR'22 | 85.24 | 57.07 | 53.58 | 55.45 | 67.19 | 82.39 | 53.70 | 49.94 | 51.96 | 63.72 |
| CURE | - | 87.05 | 58.28 | 52.10 | 55.25 | **67.60** | 86.76 | 54.92 | 49.69 | 52.48 | **65.04** |

Table 2: Performance comparisons encompass various datasets and attacks on the ResNet-18 architecture, evaluating both natural and robust performances. NRR quantifies the trade-off between natural performance and C&W attack results.

| Method | | CIFAR-100 | | | | | SVHN | | | | |
|--------|--------|-------|-------|--------|-------|-------|-------|-------|--------|-------|-------|
| | | Nat | PGD20 | PGD100 | C&W | NRR | Nat | PGD20 | PGD100 | C&W | NRR |
| AT | ICLR'18 | 57.27 | 26.66 | 26.29 | 24.89 | 34.69 | 89.21 | 51.18 | 50.35 | 48.39 | 62.74 |
| TRADES | ICML'19 | 57.94 | 29.25 | 29.10 | 25.88 | 35.77 | 90.20 | 54.49 | 54.18 | 52.09 | 66.04 |
| MART | ICLR'20 | 55.03 | 28.25 | 28.10 | 26.60 | 35.86 | 88.10 | 54.70 | 54.13 | 48.95 | 61.25 |
| FAT | ICML'20 | 61.61 | 18.35 | 17.98 | 19.31 | 29.40 | - | - | - | - | - |
| HAT | ICLR'21 | 58.73 | 27.92 | - | 24.60 | 34.67 | 92.06 | 57.35 | - | 54.77 | 68.67 |
| ST-AT | ICLR'23 | 58.44 | 30.53 | 30.39 | 26.70 | 36.65 | 90.68 | 56.35 | 56.00 | 52.75 | 66.69 |
| CURE | - | 60.72 | 30.81 | 29.82 | 27.95 | **38.27** | 92.77 | 61.56 | 58.73 | 57.34 | **70.87** |

CURE employs an innovative sparse updating technique, which not only augments the network's learning capacity but also facilitates the learning of common representation for both natural and adversarial data distributions. By retaining and selectively updating weights, we strategically avoid excessive overwriting, and ensure the preservation of previously learned natural information, thus enabling a better adaptation to adversarial data. Further, the brief revision stage in our method further improves performance by stochastically sharing stable consolidated knowledge during training. These compelling findings underscore the remarkable efficacy of CURE in striking an optimal trade-off between robustness and natural accuracy.

**Generalization and Robustness Trade-Off.** The trade-off between generalization and robustness is illustrated in Figures 1 and 5(a). Traditional approaches tend to prioritize robustness at the expense of natural accuracy. In contrast, CURE presents a balanced method that facilitates learning from standard and perturbed data. By selectively conserving weights and freeing up network capacity for learning on the current distribution, CURE ensures that natural accuracy does not significantly decline while learning adversarial samples. This approach effectively maintains a better equilibrium between robustness and generalization, offering a more desirable solution for robust training.

**Attack Strengths.** To evaluate the effectiveness of CURE under varying levels of perturbation, we conducted experiments using a PGD-20 attack with increasing perturbation strengths. The $\epsilon$ value ranges from $0.25/255$ to $8/255$. In Figure 5(b), the performance of AT and TRADES monotonically decreases and drops at higher perturbation, while MART exhibits relatively stable performance but also experiences a decline at higher perturbation levels. CURE consistently outperforms all other methods across all perturbation strengths, demonstrating its superior robustness.

**Gradients Analysis.** We gain insights into the retention and updating processes by visualizing gradients across multiple layers of the network during training. Figure 6(a) presents the mean of the gradients of the "conv2" layer within each of the four blocks of the ResNet-18 architecture. Remarkably, after the application of CURE, the gradients exhibit significantly lower magnitudes. This

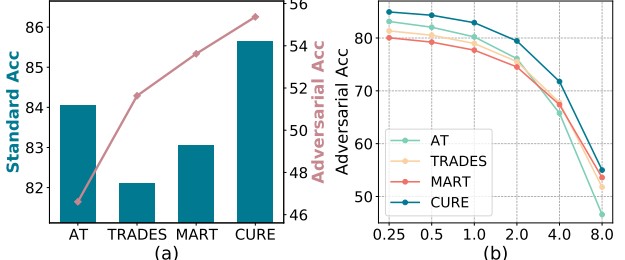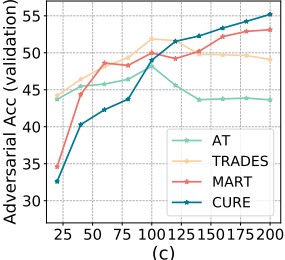

Figure 5: (a) Generalization and robustness trade-off; (b) Performance across different perturbation strengths; (c) Robust overfitting, on CIFAR-10 dataset. CURE achieves a better trade-off, and shows consistent robustness across increasing $\epsilon$ levels, while also mitigating robust overfitting.

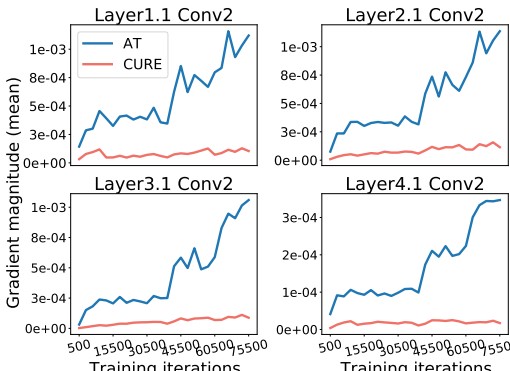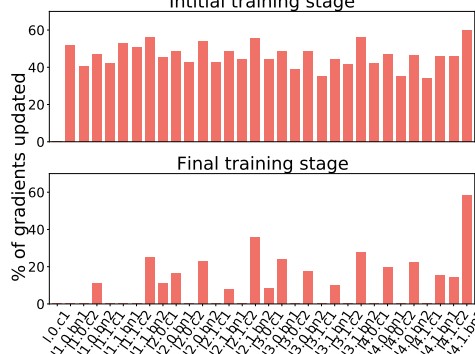

Figure 6: (a) Mean of the gradient magnitudes while training baseline and CURE; (b) Percentage of gradients updated in each layer (layer'x'.conv'y'.'weight') during initial and final phases of training.

suggests that the training mechanism entails less abrupt gradient updates at each iteration, resulting in smoother training that avoids overfitting to individual perturbed and noisy samples. Figure 6(b) illustrates the percentage of gradient updates per layer during the initial and final stages of training. In the early stages of training, a substantial number of weights undergo updates when exposed to adversarial data. However, as training progresses, the RGP metric identifies the weights that need to be fixed to prevent overwriting. Interestingly, it highlights the significance of certain middle and final layers, aligning with our empirical observations. This controlled training allows for the update of only relevant weights, maintains the knowledge learned from natural data, and adapts to acquire a more stable representation for both natural and adversarial data distributions. The strategy of sparse updating has ultimately led to enhanced trade-off and also reduced robust overfitting.

## 6 ROBUST OVERFITTING

Robustness against overfitting is a crucial concern when it comes to adversarial training. As training progresses, the network's robust performance on test data declines, in contrast to the typical behavior observed in standard training (Rice et al., 2020). This implies that the network memorizes adversarial examples instead of generalizing them to unseen examples. This phenomenon can be seen in Figure 5(c), where all methods are trained for an extended period (the same as ST). In traditional methods, such as AT or TRADES, the test accuracy for adversarial data increases to a certain point and then decreases. However, CURE effectively mitigates this overfitting issue, resulting in a consistently increasing accuracy, similar to ST.

These observations are further supported by the results presented in Table 3. CURE achieves the highest test accuracy at the end of training, indicating a $\Delta$ value of $0$, which signifies no decline in performance. It outperforms all other SOTA methods (with a negative $\Delta$), which are specifically designed to address the issue of robust overfitting. The sparse update technique in CURE not

Table 3: Robust Overfitting comparison against several methods on the CIFAR-10 dataset.

| Method | | PreActResNet-18 | | | | | | WideResNet-34-10 | | | | | |
|---|---|---|---|---|---|---|---|---|---|---|---|---|---|
| | | Natural | | | AA | | | Natural | | | AA | | |
| | | Best | Last | Δ | Best | Last | Δ | Best | Last | Δ | Best | Last | Δ |
| AT | ICLR'18 | 82.50 | 83.99 | 1.49 | 48.21 | 42.46 | -5.75 | 85.90 | 86.63 | 0.74 | 53.42 | 48.22 | -5.20 |
| TRADES | ICML'19 | 81.19 | 82.48 | 1.29 | 49.03 | 46.80 | -2.23 | 84.65 | - | - | 53.08 | - | - |
| AWP | NeurIPS'20 | 83.33 | 84.39 | 1.06 | 50.57 | 49.95 | -0.62 | 85.57 | - | - | 54.04 | - | - |
| KD | ICLR'21 | 84.51 | 85.40 | 0.89 | 49.87 | 49.72 | -0.15 | 86.81 | 87.06 | 0.25 | - | - | - |
| TE | ICLR'22 | 82.35 | 82.79 | 0.44 | 50.59 | 49.62 | -0.97 | - | - | - | - | - | - |
| IDBH | ICLR'23 | 83.96 | 84.92 | 0.97 | 50.74 | 49.99 | -0.75 | 88.61 | 89.12 | 0.52 | 55.83 | 54.01 | -1.82 |
| CURE | - | 86.54 | 86.54 | **0.00** | 50.23 | 50.23 | **0.00** | 87.05 | 87.05 | **0.00** | 52.10 | 52.10 | **0.00** |

only achieves an improved trade-off but also offers the remarkable outcome of reducing robust overfitting. The selective conservation and updation approach allows for better learning of the two data distributions and reduces overfitting to the training data. This is a significant advantage, as robust overfitting can be a major hindrance when training on complex datasets that require longer training. Unlike methods, that rely on time-consuming validation testing and early stopping techniques to achieve optimal accuracy, our approach offers a more effective training process. In summary, CURE not only enhances the network's robustness but also substantially reduces robust overfitting, effectively addressing the two primary challenges associated with adversarial training.

## 7 ROBUSTNESS AGAINST NATURAL CORRUPTIONS

DNNs are also susceptible to natural corruptions, which include distortions in input data that occur in real-world scenarios (Hendrycks & Dietterich, 2018). The inconsistent predictions under such conditions can lead to undesirable outcomes, particularly in safety-critical applications. To evaluate the robustness of our models against these corruptions, we simulate a dataset by introducing various types of corruptions (fifteen different types). Figure 7 demonstrates the efficacy of CURE in addressing various types of corruption, including blur, digital, noise, and weather. CURE achieves improved resistance to images affected by multiple corruptions. Our method focuses on striking a balance between retaining acquired knowledge from one distribution and learning a different distribution, thereby enhancing the stability and robustness of the model. These findings highlight the potential of CURE as a promising solution to enhance the robustness of neural networks in the presence of diverse corruptions.

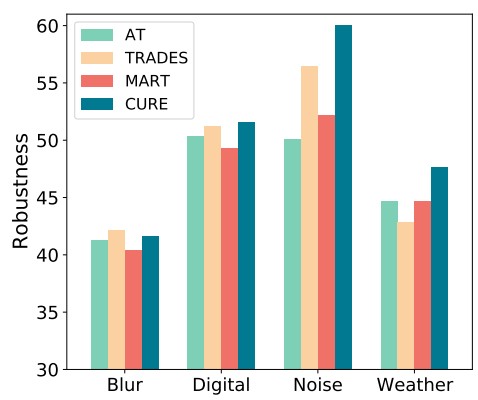

Figure 7: Robustness against various natural corruptions (at severity level 3) on CIFAR-10 dataset.

## 8 CONCLUSION

We present a novel method, denoted CURE, for adversarial training that addresses the challenge of achieving robustness without compromising generalization. Through a selective weight preservation and updating strategy, CURE enables neural networks to learn effectively from both natural and adversarial data distributions. Empirical evaluations across diverse datasets, architectures, attack types, and strengths consistently establish the superiority of CURE, showcasing its enhanced performance in terms of robustness and standard generalization. Importantly, it effectively mitigates the issue of robust overfitting, a longstanding challenge in adversarial training, while also demonstrating greater robustness to natural data corruption. In essence, our work not only sheds light on the intricate mechanisms of adversarial training but also propels the field towards an intriguing avenue of selective training schemes.

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

## A CURE ALGORITHM

---

**Algorithm 1** CURE: Conserve-Update-Revise

---

**Input:** Dataset $D$, Batch size $m$

**Initialize:** Model $f$ parameterized by $\theta$ trained on natural images $(x_{nat}, y) \sim D$

1: **while** Not Converged **do**

2:     Sample mini-batch: $(x_1, y_1), ..., (x_m, y_m) \sim D$

3:     Sample perturbation $\delta$ from a set of allowed perturbations $S$ bounded by $\epsilon$

4:     Optimize the adversarial perturbation:

$$\delta^* = \arg\max_{\delta \in S} \left[ \mathcal{L}_{cls}(x + \delta; \theta) + \alpha \mathcal{D}_{KL}\Big( f(x) || f(x + \delta) \Big) \right] \qquad \rhd \text{ Eq. 4}$$

5:     Compute both adversarial and natural losses:

$$\mathcal{L}_{adv} = \mathcal{L}_{cls}(x_{nat}; \theta) + \mathcal{D}_{KL}\Big( f(x)nat) || f(x_{adv}) \Big) \qquad \rhd \text{ Eq. 5}$$

$$\mathcal{L}_{nat} = \mathcal{L}_{cls}(x_{nat}; \theta)$$

6:     Incorporate the RGP metric:

$$\mathcal{RGP}(w) = \alpha \left\| \frac{\partial \mathcal{L}(x_{nat}; \theta)}{\partial w} \right\| + (1 - \alpha) \left\| \frac{\partial \mathcal{L}(x_{adv}; \theta)}{\partial w} \right\| \qquad \rhd \text{ Eq. 6}$$

7:     Revision Stage – Stochastically update semantic memory:

    Sample $s \sim U(0, 1)$

8:     **if** $s < r$ **then** $\theta_{rev} = d \cdot \theta_{rev} + (1 - d) \cdot \theta$          $\rhd$ Eq. 7

9:     Consistency regularizaton loss:

$$\mathcal{L}_{CR} = \mathcal{D}_{KL}\Big( p(x_{nat}; \theta_{rev}) || p(x_{nat}; \theta) \Big) + \mathcal{D}_{KL}\Big( p(x_{adv}; \theta_{rev}) || p(x_{adv}; \theta) \Big) \qquad \rhd \text{ Eq. 8}$$

10:    The overall training objective: $\mathcal{L} = \mathcal{L}_{adv} + \mathcal{L}_{CR}$

11:    Masking: compute stochastic gradients for each layer $l$

    Set the gradients lower than the $p$ percentile of RGP to 0

12:    Update the parameter $\theta$

---

## B CURE - ALTERNATIVE VERSION

We also introduce an efficient version of CURE, denoted as CURE[Eff]. CURE[Eff] does not require a pretrained model on natural images; instead, it undergoes a short warm-up phase where it is trained on natural samples before transitioning to adversarial training. This entire training process, including initial natural training and adversarial training, is completed in the same number of epochs. The results presented in Table 4 demonstrate that CURE[Eff] performs nearly as well as CURE, with only 1% lower performance.

Table 4: Performances of CURE and CURE[Eff] on ResNet-18 network on CIFAR datasets across various attacks. The second highest is underlined.

| Method | CIFAR-10 | | | | | CIFAR-100 | | | | |
|---|---|---|---|---|---|---|---|---|---|---|
| | Nat | PGD20 | AA | C&W | NRR | Nat | PGD20 | AA | C&W | NRR |
| AT | 82.78 | 51.30 | 44.63 | 49.72 | 62.12 | 57.27 | 26.66 | 23.60 | 24.89 | 34.69 |
| TRADES | 82.41 | 52.76 | 48.37 | 50.43 | 62.57 | 57.94 | 29.25 | 24.71 | 25.88 | 35.77 |
| MART | 80.70 | 54.02 | 47.49 | 49.35 | 61.24 | 55.03 | 28.25 | 25.13 | 26.60 | 35.86 |
| FAT | 87.72 | 46.69 | 43.14 | 49.66 | 63.41 | 61.61 | 18.35 | 17.98 | 19.31 | 29.40 |
| CURE | 86.76 | 54.92 | 49.69 | 52.48 | **65.04** | 60.72 | 30.81 | 25.12 | 27.95 | **38.27** |
| CURE[Eff] | 83.51 | 54.90 | 48.50 | 51.94 | 64.04 | 58.15 | 30.53 | 24.78 | 26.12 | 36.04 |

## C ADDITIONAL RESULTS

In this section, we provide additional results on more datasets and architectures.

### C.1 EFFECT OF PRETRAINING

As CURE utilizes a model trained on natural images, we also provide a comparison where the baselines utilize a trained model as well. Table 5 shows the accuracy of using pre-trained models for the two baselines. The results of the baselines do not improve solely due to the use of a pretrained

network. This demonstrates that the value of CURE lies in how we utilize the information already trained and how we retain it while learning a new data distribution.

Table 5: Performance comparisons on the CIFAR-10 dataset using pre-trained ResNet-18 architecture for all methods.

| Method | CIFAR10 | | | | CIFAR100 | | | |
|--------|---------|--------|--------|--------|----------|--------|--------|--------|
| | Nat | PGD20 | C&W | NRR | Nat | PGD20 | C&W | NRR |
| SAT | 84.37 | 43.92 | 44.14 | 57.95 | 54.33 | 20.36 | 19.96 | 29.19 |
| TRADES | 80.90 | 50.84 | 49.04 | 61.06 | 53.25 | 25.50 | 23.08 | 32.20 |
| CURE | 86.76 | 54.90 | 52.48 | **65.04** | 60.72 | 30.81 | 27.95 | **38.27** |

## C.2 GENERALIZATION TO LARGER ARCHITECTURES

To illustrate the efficacy of CURE on larger architectures, we present results for ResNet-50 and ResNet-101 in Table 6. This trend of high performance extends to larger architectures as well. Note that, for a fair comparison across methods, we maintained consistency by employing the hyperparameters of ResNet-18 for all three methods and trained them on the larger architectures.

Table 6: Performance comparisons on CIFAR10 dataset on larger ResNet architectures.

| Method | ResNet-50 | | | | ResNet-101 | | | |
|--------|-----------|--------|--------|--------|------------|--------|--------|--------|
| | Nat | PGD20 | C&W | NRR | Nat | PGD20 | C&W | NRR |
| SAT | 72.86 | 37.54 | 36.92 | 49.00 | 72.22 | 35.49 | 34.25 | 46.64 |
| TRADES | 73.06 | 40.33 | 37.56 | 49.61 | 72.28 | 39.16 | 36.69 | 48.67 |
| CURE | 74.58 | 43.85 | 41.04 | **52.94** | 73.67 | 42.80 | 38.90 | **50.91** |

## C.3 ABLATION STUDY

Through an ablation study, our aim is to provide insights into the individual contributions of two key components in CURE: *RGP* (Robust Gradient Prominence) and the *Revision* stage. The RGP plays a pivotal role in identifying weights that hold significant influence during the training process. It contributes to enhancing the network's ability to adapt to adversarial data distribution. However, to strike a balance, the revision stage is introduced, leveraging consolidated and stable information. It serves as a crucial factor in achieving equilibrium towards natural accuracy as well. The synergistic effect of including both RGP and Revision, as evident in the last row, leads to optimal performance, showcasing the effectiveness of their collective contributions.

Table 7: Ablation study on CURE components' impact on model performance trained on CIFAR10 dataset and ResNet18 architecture. NRR is calculated w.r.t. AA.

| RGP | Rev | Nat | PGD20 | C&W | AA | NRR |
|-----|-----|-----|-------|-----|-----|-----|
| ✗ | ✗ | 82.41 | 52.76 | 50.43 | 48.37 | 60.95 |
| ✓ | ✗ | 81.27 | 53.20 | 51.33 | 49.06 | 61.18 |
| ✗ | ✓ | 83.28 | 51.57 | 50.96 | 47.35 | 60.37 |
| ✓ | ✓ | **86.76** | **54.90** | **52.48** | **49.69** | **63.19** |

## C.4 NRR: ACCURACY AND ROBUSTNESS TO AUTOATTACK TRADE-OFF

Table 8: SUM and NRR metrics used to evaluate trade-off. NRR quantifies the trade-off between natural performance and AA attack results.

| Method | | WideResNet-34-10 | | | | | | ResNet-18 | | | | | |
|--------|--|------|-------|-------|-------|--------|-------|------|-------|-------|-------|--------|-------|
| | | Nat | PGD20 | AA | C&W | SUM | NRR | Nat | PGD20 | AA | C&W | SUM | NRR |
| AT | ICLR'18 | 85.17 | 55.08 | 44.04 | 52.91 | 129.21 | 58.05 | 82.78 | 51.30 | 44.63 | 49.72 | 127.41 | 57.99 |
| TRADES | ICML'19 | 84.73 | 56.82 | 52.95 | 54.29 | 137.68 | 65.17 | 82.41 | 52.76 | 48.37 | 50.43 | 130.78 | 60.95 |
| MART | ICLR'20 | 83.62 | 56.74 | 51.23 | 53.16 | 134.85 | 63.53 | 80.70 | 54.02 | 47.49 | 49.35 | 128.19 | 59.79 |
| FAT | ICML'20 | 86.60 | 49.86 | 47.48 | 49.35 | 134.08 | 61.33 | 87.72 | 46.69 | 43.14 | 49.66 | 130.86 | 57.83 |
| ST-AT | ICLR'23 | 84.92 | 57.73 | 53.54 | - | 138.46 | **65.67** | 83.10 | 54.62 | 50.50 | 51.43 | 133.60 | 62.82 |
| ACT | BMVC'20 | 87.10 | 54.77 | - | - | - | - | 84.33 | 55.83 | - | - | - | - |
| ARD | AAAI'20 | 85.18 | 53.79 | - | - | - | - | 82.84 | 51.41 | - | - | - | - |
| IAD | ICLR'22 | 83.06 | 56.17 | 52.68 | 53.99 | 135.74 | 64.47 | 80.63 | 53.84 | 50.17 | 51.60 | 130.80 | 61.85 |
| LAS-AT | CVPR'22 | 85.24 | 57.07 | 53.58 | 55.45 | 138.79 | 65.79 | 82.39 | 53.70 | 49.94 | 51.96 | 132.33 | 62.18 |
| CURE | - | 87.05 | 58.28 | 52.10 | 55.25 | **139.15** | 65.19 | 86.76 | 54.92 | 49.69 | 52.48 | **136.45** | **63.18** |

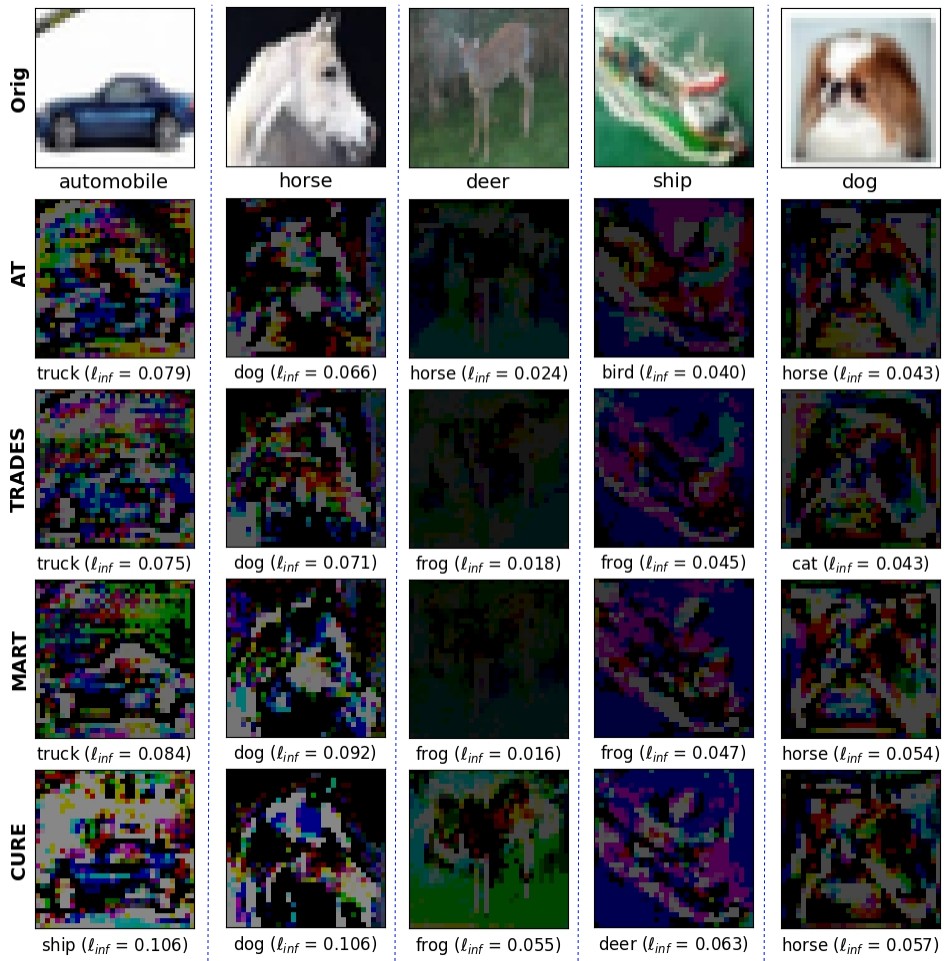

Figure 8: Minimum perturbation required to fool the models on ResNet-18 CIFAR-10 dataset.

# D    ADDITIONAL ANALYSES

## D.1    ADVERSARIAL PERTURBATION

To comprehensively evaluate the robustness of our adversarially trained models, we assess their robustness against attacks and quantify the level of difficulty in deceiving them with adversarial perturbations. This assessment is performed using the foolbox library (Rauber et al., 2017), employing the $L_{inf}$ distance metric to calculate the minimum perturbation necessary for a successful attack. To visually depict the minimum perturbations required to fool each model, we present a comprehensive visualization in Figure 8.

The visualizations provide a clear comparison of the minimum perturbations required to fool each of the robust models. The original image is shown in the first row, whereas the subsequent rows depict the minimum perturbations needed to deceive each model. In particular, the CURE model exhibits a higher level of robustness, since it requires a larger perturbation (values displayed below the image) to make an incorrect prediction compared to other methods. Furthermore, the visualizations highlight that the semantic features of the object are more preserved in CURE. In contrast to other methods, the models trained with CURE exhibit a higher level of sensitivity to perturbations. While perturbing the background or irrelevant pixels may suffice to fool the models trained with other methods, the models trained with CURE require modifications to the semantically meaningful pixels in order to be deceived. This demonstrates the effectiveness of CURE in preserving the integrity of important object features and further highlights its robustness against adversarial attacks.

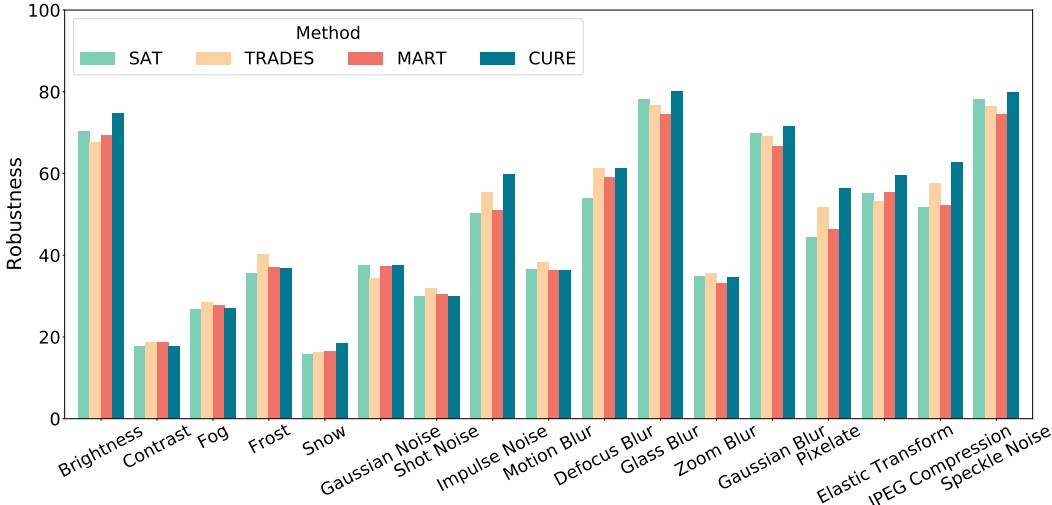

Figure 9: Robustness against 15 various natural corruptions (at severity level 3) on CIFAR-10 dataset.

## D.2 Robustness against Natural Corruptions

There are 17 distinct types of corruptions. In the main paper, we have classified them into four groups (following Hendrycks & Dietterich (2018)): Blur, Digital, Noise, and Weather, as depicted in Figure 7, to facilitate easier comparison among methods' performance. Within the Blur group, we incorporate Defocus, Glass, Motion, Gaussian and Zoom blur effects. Additionally, we explore the Digital category, introducing variations in Contrast, Elastic Transformation, JPEG Compression, and Pixelation. To simulate diverse Weather conditions, we introduce Brightness, Fog, Frost, and Snow. The Noise category encompasses Gaussian, Impulse, Speckle and Shot noise. These corruptions are systematically applied at five different severity levels, ranging from 1 (indicating less severe corruption) to 5 (representing the most severe corruption). Figure 9 illustrates the accuracy against all 15 forms of corruption.

## D.3 Gradient analysis

Figure 10 only displays the updates on convolution layers, with a higher magnitude of update observed in the middle layers. Further, in Figure 6, we observe higher updates in BatchNorm layers compared to convolutional layers. This observation underscores the significant role of batchnorm statistics in our scenario, where we initiate training from a natural model and expose it to adversarial samples with a distinct data distribution. Few works delve into exploring the influence of batch normalization in training. Notably, (Frankle et al., 2020) showcases the potential to achieve remarkably high accuracy by exclusively training the affine parameters of BatchNorm, while keeping all other parameters frozen at their original initializations.This intriguing behavior prompts further investigation into the impact of batch normalization in selective and adversarial training domains.

## E Related Works

We describe more of the SOTA methods in the extended related works. SAT (Sitawarin et al., 2021) proposes a curriculum-based loss smoothing to enhance its robustness. BS (Chen et al., 2022) introduces a new initialization strategy coupled with backward smoothing to reduce computation cost and enhance the efficiency of training. FAT (Zhang et al., 2020) investigates benign attacks that do not halt training and identifies their regularization effect on the training. ST-AT (Li et al., 2023) combines adversarial examples and collaborative examples, which exhibit significantly lower losses than their neighbors, into the training process. It formulates a joint optimization objective that minimizes the prediction discrepancies between these examples. HAT (Rade & Moosavi-Dezfooli,

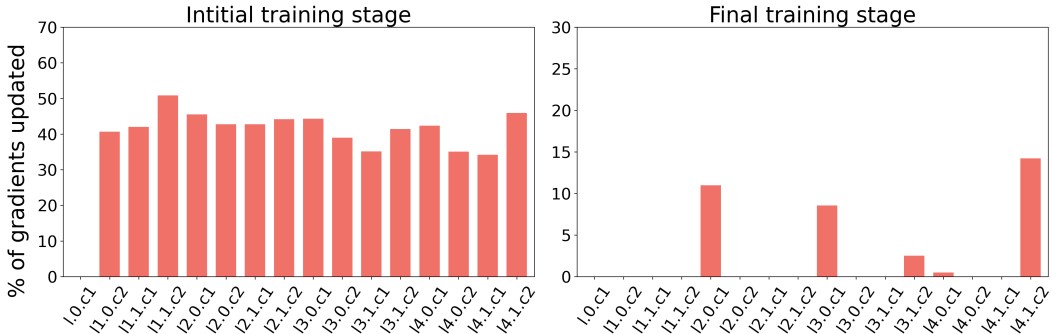

Figure 10: Percentage of gradients updated in each convolution layer (layer'x'.conv'y'.'weight') during initial, middle and final phases of training.

2021) reduces the excessive directional margin by incorporating additional wrongly labeled examples during training.

Several approaches adopt multi-network strategies during training, such as ACT (Arani et al., 2020) which concurrently trains two networks, one on natural and the other on adversarial samples. ARD (Goldblum et al., 2020) transfers knowledge from a robust teacher network to a more compact and resilient student network. Multiple models are mutually trained together in MAT (Liu et al., 2022) and self-ensemble techniques are employed in SEAT (Wang & Wang, 2021). IAD (Zhu et al., 2021) utilizes introspective adversarial distillation to improve the reliability of knowledge transfer in the presence of noisy or adversarial teacher models. LAS-AT (Jia et al., 2022) automatically generates sample-dependent adversarial attacks dynamically by utilizing an additional network.

**Robust overfitting:** In AWP (Wu et al., 2020) adversarial perturbations, computed through an optimization process to maximize the worst-case loss, are applied to the model weights, resulting in improved robustness against adversarial attacks. KD (Chen et al., 2020) proposes two approaches to mitigate robust overfitting: one using knowledge distillation and self-training to smooth the logits and the other applying stochastic weight averaging (SWA) to smooth the weights. Dong et al. (2022) explores the phenomenon of memorization and explores convergence and generalization issues through random labels. The approach integrates the temporal ensembling (TE) approach into the AT framework. Li & Spratling (2023) introduces an augmentation method called Cropshift, enhancing diversity and hardness control, and proposes an augmentation scheme called Improved Diversity and Balanced Hardness (IDBH).

## F  IMPLEMENTATION DETAILS AND METRICS

We evaluate the performance of our approach on multiple architectures, including ResNet-18 (He et al., 2016a), WideResNet-34-10 (Zagoruyko & Komodakis, 2016), and PreActResNet (He et al., 2016b). Datasets used in our study include CIFAR-10, CIFAR-100 (Krizhevsky, 2009) and SVHN. For comparison, we consider the baseline performances as reported in the respective base and comparative works. We also establish a Natural-Robusthess Ratio (NRR) metric to better measure the trade-off between natural and robust performance F.2. In the following, more details about all hyperparameters and metrics are provided.

### F.1  HYPERPARAMETERS

For our method, all models are trained using the SGD optimizer with a momentum of $0.9$. The augmentations include basic random crop and random flip operations. Projected Gradient Descent (PGD) is used to generate adversarial images. For adversarial training, PGD with step $10$ is considered with perturbation strength $\epsilon = 8$ and step size $\epsilon/4$. Table 9 tabulates the other hyperparameters used in our method. Our revision stage occurs stochastically with only a $20\%$ probability. Moreover, as

Table 9: Hyperparameters used for CURE. The learning rate is $0.1$, the number of epochs is $200$ and the weight decay is $5e^{-3}$. The revision rate $r$ and decay factor $d$ for the revision stage are set to $0.2$ and $0.999$ for all the experiments.

| Architecture | Dataset | $p$ | $\alpha$ | $\gamma$ |
|---|---|---|---|---|
| ResNet-18 | CIFAR-10 | 30 | 0.1 | 1.0 |
| | CIFAR-100 | 30 | 0.1 | 1.0 |
| | SVHN | 30 | 0.2 | 1.0 |
| WideResNet-34-10 | CIFAR-10 | 50 | 0.3 | 0.5 |
| | CIFAR-100 | 50 | 0.3 | 0.5 |
| PreActResNet-18 | CIFAR-10 | 20 | 0.2 | 1.0 |

Table 10: Hyperparameters sensitivity analysis in CURE. The hyperparameters remain stable across various settings and datasets. Among them, "$p$" and "$\alpha$" are important and are intuitive, as they determining the trade-off between natural and adversarial accuracy.

| | Sensitivity to $\gamma$ | | | | | Sensitivity to $p$ | | | | | Sensitivity to $\alpha$ | | | |
|---|---|---|---|---|---|---|---|---|---|---|---|---|---|---|
| $p$ | $\alpha$ | $\gamma$ | Nat | Rob | $p$ | $\alpha$ | $\gamma$ | Nat | Rob | $p$ | $\alpha$ | $\gamma$ | Nat | Rob |
| 30 | 0.1 | 1.0 | 86.76 | 54.92 | 30 | 0.1 | 1.0 | 86.76 | 54.92 | 30 | 0.1 | 1.0 | 86.76 | 54.92 |
| | | 0.5 | 86.25 | 53.13 | 10 | | | 85.15 | 55.65 | | 0.2 | | 86.95 | 54.17 |
| | | | | | | | | | | | 0.5 | | 88.52 | 53.67 |

training advances, we gradually decay the revision rate to minimize revisions when the network stabilizes.

In the evaluation setting, we examined several white-box attacks and a hybrid attack that combines both white- and black-box methods. White-box attacks have full access to the model parameters and are limited by a maximum perturbation of $\epsilon$. On the other hand, black box attacks are created by attacking a surrogate model that has the same architecture as the targeted model, using natural images as input.

PGD (Projected Gradient Descent) is considered to be one of the strongest first-order attacks, as it is able to find the maximum possible perturbation that will cause the model to misclassify an input. The PGD attack starts by generating a random initial perturbation and then repeatedly takes gradient steps in the direction that increases the model's classification error, until the maximum perturbation limit is reached. This is done by projecting the perturbation back onto the $\epsilon$-ball after each step, which ensures that the perturbation does not exceed the maximum allowed value. The PGD attack can be customized to use different values of step size and number of steps. In the evaluation discussed in the previous information, we are using three variations of the step size ($10$, $20$, and $50$) and various perturbation strengths of $\epsilon$. C&W (Carlini & Wagner, 2017) is also known to be one of the strongest white-box attacks. The C&W attack is based on an optimization problem that tries to minimize the L2 distance between the original input and the adversarial example, while also maximizing the model's classification error. AutoAttack (AA) (Croce & Hein, 2020b), is an automated attack method that is designed to be both reliable and strong. AA consists of white-box attacks: APGD-CE, APGD-DLR (Croce & Hein, 2020b), FAB (Croce & Hein, 2020a) and black box attack: Square (Andriushchenko et al., 2020). The APGD-CE (AutoAttack with PGD and Cross-Entropy loss) and APGD-DLR (AutoAttack with PGD and Data-Limited-query loss) methods are based on the Projected Gradient Descent (PGD) attack, and they use different loss functions (Cross-Entropy and Data-Limited-query loss, respectively) to optimize the perturbation. The square attack is a black-box attack that leverages the ability of the model to classify images of squares and rectangles in order to generate adversarial examples. The 'Fast and Accurate Black-Box Attack' (FAB) is also included in AutoAttack.

**Sensitivity** : The hyperparameters are stable across settings and datasets and also complement each other. Therefore, CURE does not require extensive fine-tuning across different datasets and settings. Some hyperparameters, like the revision rate ($r$) and decay factor ($d$) for the revision stage, are set universally ($r = 0.2$, $d = 0.999$) across all experiments. The revision rate ($r$) determines the

stochasticity of the revision stage, with a $20\%$ probability of occurrence. This is more important during the initial stage of training and controlled reduction in revisions occurs as training progresses, as the network stabilizes.The hyperparameter $\gamma$, which is associated with the consistency regularization term, is generally stable and is set to $1.0$.

The main hyperparameters of interest are "$p$" and "$\alpha$", which are quite intuitive. The parameter "$\alpha$" influences the importance given to the adversarial distribution in the learning process. Higher values of $\alpha$ emphasize natural accuracy. Setting $\alpha$ to a lower value, such as $0.1$ or $0.2$, is preferable, especially when the network is pre-trained in the standard setting. The $p$ parameter is associated with estimating the percentile in gradients below which the gradients are set to zero. The choice of $p$ influences the balance between retaining previous knowledge and learning new information. More details are provided in Table 10

**Computation cost and limitations:** In terms of computation cost, CURE utilizes a single network for inference, similar to other methods. During the training process, an additional pass is performed to calculate and choose the relevant weights, and during the revision stage, a stochastic momentum update of the training model is utilized.

## F.2 NATURAL-ROBUSTNESS RATIO

In the field of deep neural networks, it is often crucial to strike a balance between two fundamental aspects of model performance: natural accuracy and robust accuracy. Natural accuracy refers to a model's ability to perform well on standard, unaltered data that closely resembles the training data distribution. On the other hand, robust accuracy assesses the model's performance when exposed to perturbed or adversarial data, which may deviate significantly from the training distribution.

The *NRR (Natural-Robustness Ratio)* metric is introduced as a means to quantitatively measure this trade-off between natural accuracy and robust accuracy. It considers both aspects of model performance when dealing with standard and perturbed data.

$$NRR = \frac{2 \times Natural\,Accuracy \times Robust\,Accuracy}{Natural\,Accuracy + Robust\,Accuracy} \tag{9}$$

The NRR metric combines these two accuracy measurements into a single value. A high NRR value indicates that the model excels in both natural and robust accuracy simultaneously, achieving a well-balanced trade-off. Conversely, a lower NRR value suggests that the model might prioritize one aspect (natural or robust accuracy) over the other, potentially sacrificing performance in the less prioritized dimension.

## F.3 CKA

Central Kernel Alignment (CKA) is a method for measuring the similarity between two sets of features, such as the features of a neural network's intermediate layers. It is commonly used to analyze the representations learned by neural networks and to understand how different architectures or training methods affect the representations.

$$CKA(F_1, F_2) = \frac{K_1 \cdot K_2}{|K_1|_F \cdot |K_2|_F} \tag{10}$$

where $F_1$ and $F_2$ are the two sets of features being compared, $K_1$ and $K_2$ are the corresponding kernel matrices, and $|\cdot|_F$ denotes the Frobenius norm.

CKA is calculated by first computing the dot product of the two sets of features, then squaring the result, and finally normalizing it by the product of the norms of the two sets of features. The result is a scalar value between -1 and 1, where a value close to 1 indicates high alignment, a value close to -1 indicates low alignment, and a value close to 0 indicates no alignment.

## G   EMPIRICAL ANALYSIS DETAILS

Balancing the trade-off between generalization and robustness remains an enduring challenge in the field of AT, exacerbated by the intricacies of robust overfitting. Furthermore, while standard test

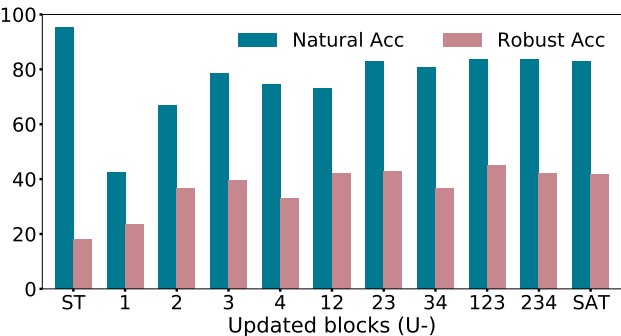

Figure 11: Standard generalization and robustness of different blocks of ResNet-18 on CIFAR-10 dataset without re-initializing layers.

accuracy typically exhibits an upward trajectory or plateaus during ST, it experiences a decline in the adversarial setting, signaling underfitting. And the decrease in accuracy on clean images suggests memorization. Fixing the parameters of a naturally trained model before adversarial training can provide insights into the learning behavior of the network, which is crucial for achieving a balance between generalization and robustness. Hence, we advocate for incorporating layer-wise properties into developing training schemes.

Re-initialization, or the process of randomly initializing the weights of a neural network during training, can provide detailed insights into what each layer of the network is learning. Especially, when training data on different distributions, re-initialization can prove important to learn how the network performs on both distributions. Re-initialization has been explored in standard training (transfer learning and fine-tuning) but not in adversarial setting. Here, we investigate the layer-wise properties of a network through the lens of retention and updation.

**Setup:** We use a pre-trained model (on the original CIFAR-10 dataset) as the base network. We freeze the parameters and perform AT(Madry et al., 2018) on CIFAR-10 dataset. The notation U-b represents the update of block "b" while keeping the rest of the network frozen. We create 12 different combinations (U-1, U-2, U-3, U-4, U-12, U-23, U-34, U-123, U-234). The example of U-23 is shown in Figure2(a).

**Adversarial Robustness:** In Figure 2(b)., the "ST" and "SAT" represents the accuracy of the ST and AT (Madry et al., 2018). The label "1" ("U-1") represents the freezing of all layers except the first block, which is trained and updated on adversarial images. Similarly, "23" represents freezing block 1, 4, and the final layer 5, while re-initializing and updating the weights of the second and third block of the network. Among the models trained with combinations of layers, freezing the first and last layers and updating the weights of the second and third proves beneficial in increasing both natural and adversarial performance (compared to the AT baseline). This shows that updating the parameters of the whole network is not an optimal approach to train the two data distributions.

Figure 11 displays a similar analysis, albeit without re-initializing the layers; in other words, the blocks are solely fine-tuned on the adversarial data. While the results show slightly higher performance in certain settings, the overall pattern remains consistent with the re-initialization analysis.

**Representation Alignment:** Figure 3 shows the similarity between natural and robust representations for each of the models. For the nine models, trained with different retention and update strategies, a pattern of "block structures" is observed. Block structures indicate a high level of intra-layer similarity and lower inter-layer similarity. For instance, in the U-1 model, the first block displays considerable dissimilarity with the fourth block, while the last three blocks exhibit higher similarity. Notably, the block structure becomes less pronounced in later layers. Moreover, models that selectively update only the final layers demonstrate more similarity in the representations (e.g., U-4, U-34, U-234). The representation similarity between different layers in the DNNs trained by CURE is visualized using the CKA metric in Figure 12. Compared to standard training (ST) and adversarial training (AT), CURE lies in the middle, leaning towards enhanced robust accuracy. It demonstrates a more balanced and nuanced performance, with an overall superior trade-off.

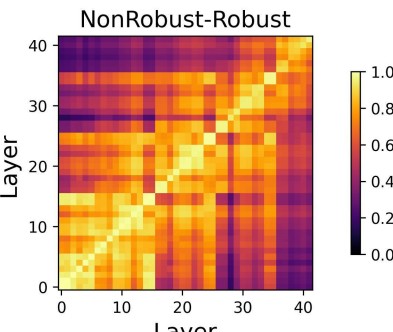

Figure 12: Representation similarity between robust and nob-robust features in ResNet-18 trained on the CIFAR-10 dataset using CURE.

**Robust Overfitting:** Robust overfitting is a common issue in adversarial training, where test accuracy tends to decrease while training accuracy increases or plateaus, indicating overfitting. Figure 4 displays the test accuracy curves for all eight models. In the first graph, freezing layer1 and layer2 individually reduces overfitting but leads to a decrease in overall accuracy. In the second graph, updating the last two layers (U-34) exhibits overfitting behavior, while updating the first two layers (U-12) and the middle two layers (U-23) perform better.

