# OpenReview forum: "Conserve-Update-Revise to Cure Generalization and Robustness Trade-off in Adversarial Training"
_ICLR.cc/2024/Conference — ICLR 2024 poster_

### Official Review · Reviewer_PH92 · 2023-10-28

**Soundness:** 2 fair
**Presentation:** 3 good
**Contribution:** 2 fair
**Rating:** 5
**Confidence:** 3

**Summary:**

This paper proposes a new method to improve the trade-off between robustness and generalization in adversarial training.
First, this paper investigates the difference between adversarial and clean representations layer-wisely,
and next, it investigates overfitting in adversarial training when parameters of only some layers are selectively updated.
Based on the observation that some layers tend to suffer from overfitting, this paper proposes CURE that leverages a gradient prominence criterion to perform selective conservation, updating, and revision of weights.
To evaluate the trade-off, this paper establishes a new metric: Natural-Robustness Ratio which is calculated by using accuracy against C&W and natural accuracy.
CURE is evaluated in terms of this metric, robustness against several attacks including AutoAttack, and robustness against natural corruption.

**Strengths:**

- This paper addresses an important problem in adversarial training: Overfitting and the trade-off between natural accuracy and robustness.
- The detailed investigation of layer-wise learning phenomena in adversarial learning is novel and provides interesting insights.
Revealed layer-wise properties might inspire researchers in this area and might cause new defense methods.
- Gradient-based selective update for adversarial training is a new and interesting idea.
The figure of gradients in the training (Fig. 6) intuitively shows how the proposed method works by using the information of gradients well.
- CURE is evaluated by using various attacks and architectures. However, baselines are not consistent and results might be cherry-picked.

**Weaknesses:**

- This paper lacks an ablation study. Although layer-wise analyses are interesting, the proposed method contains several components besides selective updating. How is the performance if we use only RGP?
If it is not good, are equations (4), (5), (7), and (8) relevant to the layer-wise analysis?
If other parts than the selective updating contribute to the performance, the layer-wise analysis may not be very worthwhile.

- This paper does not present a fair and honest evaluation and presentation of the results of the experiments.
The trade-off metric is intentionally designed to make the proposed method look overly good. Is there a rational explanation as to why the C&W is used in the evaluation of trade-off, even though AutoAttack performs better than C&W in terms of attack success rate? I suspect that C&W is chosen because the numbers of metrics are not better in the case of AutoAttack. In fact, robustness against AutoAttack of the proposed method is not always greater than baselines.
Additionally, the vertical and horizontal scales in Figure 1 are not aligned, which can be misleading.
Natural corruptions are selectively used from CIFAR10C. Their results may be cherry-picked. I would like to see the results against all natural corruptions in CIFAR10C. Baseline methods are not consistent over expreiments.
Why does Table 1 not contain the result of HAT, and does Table 2 not contain ACT, ARD, LAD, and LAS-AT?

- The boundary between the proposed and existing methods is described ambiguously. Eqs. (4) and (5) seem to be an objective function of TRADES. Why are these equations written in the section of the proposed method?
Additionally, SMU seems to be exponential moving average (EMA) with a stochastic parameter. SEAT (Wang & Wang, 2021) and other recent methods also use EMA, which is sometimes called weight averaging. Unlike them, the proposed method uses averaged parameters in the regularization term. I would like to see the comparison between averaging weights directly and using averaged weights for regularization.
- Minor issues
    - [a] might be related work that addresses the trade-off and focuses on the difference between the representations of clean data and adversarial examples. The method in [a] outperforms LAS-AT in terms of the trade-off. Since it seems to be concurrent work, I think that it is not necessary to compare.
    [a] Suzuki S et al. "Adversarial Finetuning with Latent Representation Constraint to Mitigate Accuracy-Robustness Tradeoff." ICCV 2023.

**Questions:**

- How is the performance if we use only RGP? If it is not good, are equations (4), (5), (7), and (8) relevant to the layer-wise analysis?
- Is there a rational explanation as to why the C&W is used in the evaluation of trade-off? Is there any reasonable explanation for Natural-Robusthess Ratio? Why is eq.(9) suit to evaluate the trade-off?
- How does NRR become if using AutoAttack?
- What is the difference between TRADES and eqs.(4) and (5)?
- Why does Table 1 not contain the result of HAT, and does Table 2 not contain ACT, ARD, LAD, LAS-AT?

---

> ### Author Response · Authors · 2023-11-19
> **Response to Reviewer PH92 - 1**
>
> Thank you for finding our work insightful and providing valuable feedback. Below, we provide our responses to your questions.
>
> > ablation study
>
> We add the ablation results in Table 7 in the Appendix. The RGP metric significantly identifies influential weights during training, prioritizing the enhancement of the network's capability to handle adversarial challenges. To maintain a balance, the revision stage utilizes consolidated and stable information, serving as a vital factor in achieving a balance, including improved performance in natural accuracy.
> Also, revision stage is more important initially, to avoid drastic shift from natural to adversarial during training. As training progresses, the revision rate is systematically reduced to minimize revisions, particularly when the network stabilizes. The combined impact of both RGP and Revision, as observed in the final row, results in optimal performance, highlighting the effectiveness of their collaborative contributions.
>
> > Is there a rational explanation as to why the C&W is used in the evaluation of trade-off? Is there any reasonable explanation for Natural-Robusthess Ratio? Why is eq.(9) suit to evaluate the trade-off?
>
> We appreciate the reviewer's diligence in evaluating our work. We respectfully disagree with the assertion that our paper lacks fair and honest evaluation.
>
> **NRR** :  Many adversarial studies predominantly show accuracy metrics, and some approaches have higher robust accuracy even if it comes at the cost of natural accuracy. Our objective was to not just improve adversarial accuracy, but to achieve better trade-off between natural and adversarial generalization. This we wanted to identify a metric that effectively captures and illustrates this trade-off in our method. Consequently, we conducted a thorough search for such a metric. In our exploration of trade-off evaluation metrics, we observed a prevalent use of the "SUM" metric in many papers (including the suggested paper [a] in the review). However, it became evident that relying solely on a SUM metric is problematic. Consider the scenario where 50% natural accuracy and 50% adversarial accuracy are treated worse than 90% natural accuracy and 11% adversarial accuracy  (even at chance level). This simplistic SUM approach fails to provide a nuanced evaluation of the delicate balance between both accuracies, rendering it ineffective in capturing the essence of the trade-off.
>
> To address this limitation, we propose the Natural-Robustness Ratio (NRR) metric (Eq. 9), drawing inspiration from the F1 score commonly used in object detection (to balance precision and recall). This ratio offers a more meaningful assessment by considering the harmonious interplay between natural and adversarial accuracy, avoiding the oversimplification inherent in the SUM metric. We are open to further suggestions or refinements to our evaluation metric,  If the reviewer has specific recommendations for alternative metrics, we would like to discuss more and explore.
>
> **C&W attack** : Our choice of the C&W attack for the evaluation of trade-off is based on common practices in the field and the availability of consistent metrics across different papers. C&W, which is also known to be one of the strongest white-box attacks, was an attack we saw in papers from both categories : overfitting and generalization and across datasets, including SVHN. Hence we used that for NRR.
>
> To address the concern raised, we have included an additional Table 8 in the Appendix that presents the Natural-Robustness Ratio (NRR) with AutoAttack. While we outperform in ResNet18, in WideResNet, we are close. However, as observed, the weakness lies in the metric. For example, concerning ST-AT, CURE exhibits a more significant increase in natural accuracy (84.92->87.05) and a smaller decrease in robust accuracy (53.54->52.10). However, the NRR is still higher for ST-AT. We explored various metrics to capture the trade-off, and we are open to further suggestions from the reviewer.
>
> > Additionally, the vertical and horizontal scales in Figure 1 are not aligned, which can be misleading.
>
> We employed different scales to enhance clarity and prevent crowding of baselines at the same point. in Figure 1. We have kept both on a linear scale. We have generated an additional version of Figure 1 with the same scales for both axes, and we have included it in Figure 13 of Appendix.

---

> > ### Author Response · Authors · 2023-11-19
> > **Response to Reviewer PH92 - 2**
> >
> > > Natural corruptions are selectively used from CIFAR10C. Their results may be cherry-picked. I would like to see the results against all natural corruptions in CIFAR10C.
> >
> > I believe there might be a misunderstanding. We did not selectively choose four corruptions in Figure 7; instead, we grouped various corruptions into four categories, following the approach used in [1].
> >
> > There are two ways to report these results: either present all corruptions individually in one table or group similar corruptions into categories and provide an average over these categories.
> > Our categorization is as follows (which is followed by [1]):
> > Blur group: Defocus, Glass, Motion, Gaussian, and Zoom blur effects.
> > Digital category: Contrast, Elastic Transformation, JPEG Compression, and Pixelation.
> > Weather conditions: Brightness, Fog, Frost, and Snow.
> > Noise category: Gaussian, Impulse, Speckle, and Shot noise.
> >
> > In the main paper, we systematically organized these corruptions into the four mentioned groups for a clear and comprehensive plot. Additionally, we provide detailed results for each corruption in the appendix in Figure 9, to ensure transparency and completeness.
> >
> > [1] Dan Hendrycks and Thomas Dietterich. Benchmarking neural network robustness to common corruptions and perturbations. In International Conference on Learning Representations, 2018
> >
> > > What is the difference between TRADES and eqs.(4) and (5)? Unlike SEAT, the proposed method uses averaged parameters in the regularization term. I would like to see the comparison between averaging weights directly and using averaged weights for regularization.
> >
> > We appreciate the reviewer's observation and would like to clarify that Eqs. (4) and (5) represent specific components within the broader context of our proposed method, CURE. The objective was added to improve comprehension as well.
> > While these equations incorporate aspects of the regularization objective commonly used in adversarial training, In CURE, the key value lies in the integration of the Regularization on Gradients Perturbation (RGP) metric and the strategic weight preservation and updating mechanism based on this metric. The RGP metric plays a crucial role in identifying and quantifying the importance of individual weights, guiding the selective preservation and updating process during training.
> >
> > The SEAT method primarily focuses on tracking the weight states of models throughout the training process, incorporating a self-ensemble strategy by averaging the weights of historical models. In CURE, the revision stage is one part of the design. CURE involves the use of an exponential moving average (EMA) to average the weights of the model into a revision model during the revision stage. This consolidated model is then utilized for transferring consolidated knowledge to the network
> >
> > Further, the revision model in CURE undergoes updates stochastically, triggered by a sampled scalar below a decay rate of 0.2. This stochasticity ensures that the revision stage occurs with a 20% probability, and as training progresses, the revision rate is systematically reduced to minimize revisions, particularly when the network stabilizes. As we start from naturally trained model, revision stage is more useful initially, as the shift is more drastic.
> > We also provide the finally accuracy on the averaged model below.
> >
> > | CIFAR10-ResNet18 |  |  |
> > |---|---|---|
> > |  | Nat | Rob |
> > | CURE | 86.76 | 54.92 |
> > | CURE-Revision model | 85.03 | 50.47 |
> >
> > If there are any specific aspects or details that we might have overlooked or misunderstood, kindly provide additional clarification.
> >
> > > Why does Table 1 not contain the result of HAT, and does Table 2 not contain ACT, ARD, LAD, LAS-AT
> >
> > The baseline methods presented in Table 1 and Table 2 were sourced from the respective original papers. Maintaining consistency across all architectures and datasets would significantly limit the number of methods included in the tables, resulting in a sparse representation with only 3-4 methods.
> >
> > To maintain fairness, the results for a specific dataset and architecture were extracted from the original works, as these papers fine-tuned their methods to achieve optimal performance. Unfortunately, for certain older methods like ACT and ARD, we encountered challenges in finding results for specific architecture-dataset combinations, and the best parameter settings were not always explicitly provided in their papers.
> >
> > This practice aligns with the conventions in adversarial literature, where reporting available and representative results is seen due to the lack of a comprehensive baseline and discrepancies in the literature.

---

> > > ### Comment · Reviewer_PH92 · 2023-11-22
> > > **Thank you for the feedback!**
> > >
> > > Thank you for the reply, and some of my concerns are addressed. However, new concerns have arisen. I'm leaning towards reject.
> > >
> > > > ablation study
> > >
> > > I like a new result of Table 7! However, I have new questions about this result as mentioned below. Additionally, I would like to see the robust accuracy against AutoAttack.
> > >
> > > > We have generated an additional version of Figure 1 with the same scales for both axes, and we have included it in Figure 13 of Appendix.
> > >
> > > My comment may not have been clear. My comment was to show Fig. 1 as a square grid on the same scale. x-axis and y-axis in Fig. 13 are still not on the same scale.
> > >
> > > > However, it became evident that relying solely on a SUM metric is problematic.
> > >
> > > I agree that there is a problem with the sum metric but do not think that NRR is the perfect metric. I think that a sum metric is useful enough to be listed with NRR in tables.
> > >
> > > >  C&W, which is also known to be one of the strongest white-box attacks
> > >
> > > I agree that C&W tends to more strong than PGD, but AutoAttack is the strongest. I think NRR with AutoAttack (table 8) should be located in the main paper.
> > >
> > > > I believe there might be a misunderstanding. We did not selectively choose four corruptions in Figure 7; instead, we grouped various corruptions into four categories, following the approach used in [1].
> > >
> > > I am sorry for misunderstanding the results of natural corruption. A new figure is more informative, and it still shows the effectiveness of the proposed method.
> > >
> > > > We appreciate the reviewer's observation and would like to clarify that Eqs. (4) and (5) represent specific components within the broader context of our proposed method, CURE.
> > >
> > > In reply, it appears that the only difference between the methods is the terminology: the RGP does not depend on TRADES losses (i.e., the naive adversarial loss can be used), and eqs (4) and (5) only indicate a combination of TRADES and RGP. In fact, the adversarial loss of line 5 of the algorithm can be exchanged as $L_{cls}(x_{adv})$. On the other hand, the baselines include those using naive adversarial losses. For example, at least, AWP is not used with TRADES in Table 3. This difference makes the performance comparison a bit unfair.
> > >
> > > Furthermore, looking at the results of the ablation study, the only RGP appears to be worse than TRADES. In the experiment of
> > > only Revision stage in Table 7, what was used for losses? If Rev+TRADES reaches the same performance as Rev+RGP, then the masking of gradients in RGP is meaningless.
> > >
> > > > The baseline methods presented in Table 1 and Table 2 were sourced from the respective original papers.
> > >
> > > I think the experimental results reproduced by authors are better than those reprinted from the original papers because authors can directly design the experiments to unify the experimental conditions. Therefore, the absence of results in the original paper is not a reason not to include comparative results in this paper.

---

> > > > ### Author Response · Authors · 2023-11-22
> > > > **Response to the new concerns/questions (2/3)**
> > > >
> > > > > In reply, it appears that the only difference between the methods is the terminology: the RGP does not depend on TRADES losses (i.e., the naive adversarial loss can be used), and eqs (4) and (5) only indicate a combination of TRADES and RGP. In fact, the adversarial loss of line 5 of the algorithm can be exchanged as Lcls. On the other hand, the baselines include those using naive adversarial losses. For example, at least, AWP is not used with TRADES in Table 3. This difference makes the performance comparison a bit unfair.
> > > >
> > > > The literature on adversarial training (AT) predominantly employs either the Madry or TRADES objective functions. The rationale is that AT often involves training with adversarial losses and, in addition, may include a regularization term for natural images.  The majority of compared methods, integrate the regularization objective.
> > > >
> > > > For example, MART employs the TRADES loss but introduces extra regularization for misclassified samples, while HAT leverages the TRADES objective with the inclusion of additional wrongly labeled examples during training. Many distillation and collaborative training methods (ACT, ARD)  which have additional teacher networks and external natural models to provide additional regularization. IDBH utilizes a new augmentation during training. CURE doesn't have to compare against these as they have additional memory and resources during training, but each method is specifically designed to address the problem in its unique way, and we should avoid oversimplifying the diversity of approaches within the AT landscape.
> > > >
> > > > CURE represents a selective adversarial training approach, distinguishing itself by dynamically updating weights in a selective manner. This innovative method aims to effectively learn from both natural and adversarial data, providing a nuanced and flexible strategy in comparison to traditional adversarial training (AT) techniques. And notably, CURE outperforms TRADES on all settings.
> > > >
> > > > In our comparisons, when a method distinguishes between Madry and TRADES, we prioritize reporting the TRADES variant for consistency. For instance, in the case of LAS-AT and FAT, we present results based on the TRADES variant.
> > > >
> > > > The primary goal of CURE was to enhance the trade-off. However, during CURE training the accuracy increases and saturates, in contrast to other adversarial training (AT) techniques that exhibit a declining trend over extended training periods (Fig. 5c).Thus selective training of CURE also contributes to mitigating robust overfitting.  Some AT works are only focused on trade-off and others on overfitting. For evaluation of robust overfitting behavior, we required best and last accuracy results. Original AWP paper did not have the best and last result comparison for the TRADES variant, and hence we based the comparison off the latest comprehensive evaluation from the IDBH [ICLR 23] paper for Table 3 results.

---

> ### Author Response · Authors · 2023-11-22
> **Response to the new concerns/questions (1/3)**
>
> > I like a new result of Table 7! ... Additionally, I would like to see the robust accuracy against AutoAttack.
>
> > ... In the experiment of only Revision stage in Table 7, what was used for losses? If Rev+TRADES reaches the same performance as Rev+RGP, then the masking of gradients in RGP is meaningless.
>
> We have incorporated the AutoAttack results into Table 7, as per the reviewer’s suggestion.
>
> It seems there might be a misunderstanding regarding the ablation study. When incorporating only RGP, since we start from a naturally trained model. The metric identifies layers that can effectively learn the new adversarial distribution. Consequently, when only RGP is added, adversarial accuracy increases compared to the first row.
>
> In the revision stage only, the objective is to consolidate information gradually, preventing the network from forgetting natural accuracy while learning the new data distribution. Therefore, the revision-stage-only scenario, which follows the objective of equation [5], exhibits higher natural accuracy than the first row, but the robust accuracy decreases. CURE leverages both selective updating and consolidation, aiming to achieve a better trade-off between natural and robust accuracy, and our extensive empirical study demonstrates this optimal trade-off.
>
> | RGP | Rev  | Nat | PGD20 | C&W  | AA | NRR |
> |-------|-------|-----|---------|---------|----|-------|
> | ✗ | ✗ | 82.41 | 52.76 | 50.43 | 48.37 | 60.95 |
> | ✓ | ✗ | 81.27 | 53.20 | 51.33 | 49.06  | 61.18 |
> | ✗ | ✓ | 83.28 | 51.57 | 50.96 | 47.35  | 60.37 |
> | ✓ | ✓ | **86.76** | **54.90** | **52.48** | **49.69** | **63.19** |
>
> > My comment may not have been clear. My comment was to show Fig. 1 as a square grid on the same scale. x-axis and y-axis in Fig. 13 are still not on the same scale.
>
> Figure 13 has been updated to align with your suggestion.  However, we are a bit confused regarding this. Could you please provide more details to help us understand the specific concern?
>   - The natural accuracy consistently falls within the range of - [80-90]
>   - The adversarial accuracy scale spans a lower range ->  [45-60]
>
> Figure 1 is what would be generated when these values are provided to any plotting tool, as by default they inherently adjust the scales based on the provided data for a comprehensive visualization.
> However, to generate Figure 13, we have fixed the scales to accommodate the request. Typically, when visualizing factors with varying ranges, such scale alignment is not a common practice. We are eager to understand the reasoning behind your request.
>
> > I agree that there is a problem with the sum metric but do not think that NRR is the perfect metric. I think that a sum metric is useful enough to be listed with NRR in tables.
>
> Both SUM and NRR metrics have been incorporated into Table 8, as per reviewer’s suggestion.
>
> We acknowledge and have also stated in the previous rebuttal about the NRR not being the perfect  metric. However, we believe it still holds more value than the SUM metric in portraying the trade-off between natural and robust accuracy.
> *Could you elaborate on how the SUM metric effectively communicates the trade-off compared to NRR?*
>
> It's also crucial to address the reviewer's claim that the trade-off metric is intentionally designed to favor CURE. CURE excels in the SUM metric (Table8), but we still believe that SUM cannot capture the trade-off well at all. A high SUM alone does not necessarily signify an optimal trade-off; it could result from emphasizing a single accuracy metric.
>
> Our primary objective is to enhance the generalization trade-off, emphasizing a balanced improvement across both natural and robust accuracy metrics. Our approach considers the absolute values of accuracy against various attacks, and we have introduced the NRR metric (inspired from F1 score) as an additional measure to provide a more comprehensive view of the trade-off landscape.
>
> > I agree that C&W tends to more strong than PGD, but AutoAttack is the strongest. I think NRR with AutoAttack (table 8) should be located in the main paper.
>
> To ensure consistency in the paper for readers, we established the NRR using C&W (similar to many literature sources that demonstrate the use of the SUM metric only with PGD20) present in all tables, which cover various architecture-dataset combinations. Particularly regarding Tables 1 and 2, as the second table does not include the AA.
>
> Nevertheless, we are open to accommodating the reviewer's feedback by swapping the positions of these two tables and displaying the NRR on AA in the main paper.
>
> > I am sorry for misunderstanding the results of natural corruption. A new figure is more informative, and it still shows the effectiveness of the proposed method.
>
> We are glad to hear it.

---

> ### Author Response · Authors · 2023-11-22
> **Response to the new concerns/questions (3/3)**
>
> > I think the experimental results reproduced by authors are better than those reprinted from the original papers because authors can directly design the experiments to unify the experimental conditions. Therefore, the absence of results in the original paper is not a reason not to include comparative results in this paper.
>
> Our reporting approach involves two main considerations. Firstly, we include accuracy results from the original papers or comparative survey works. This choice is driven by the fact that the authors of the respective methods have fine-tuned their models for specific datasets and architectures, representing the optimal performance achievable under their established conditions. Furthermore, the essence of scientific publication is to facilitate the reusability of results.
>
> Several studies opt to train their foundational methods with the hyperparameters they deem optimal for their method, considering it a fair practice. It's important to note that each method has a specific set of parameters that yield the best performance. For instance, methods like Madry and TRADES incorporate early stopping to prevent overfitting, which can lead to a reduction in the final results. We conducted training for these methods, and while the results were notably lower than the reported original results, we chose to include the latter in our tables. This decision was made to ensure consistency and transparency, acknowledging the varied approaches in the literature and providing a clear basis for comparison.
> Notably, numerous papers lack results for specific architecture-dataset combinations. The generation of reliable results requires meticulous fine-tuning to find the best parameters for this combination before comparing, and factors like seed values also influence the outcomes. Recognizing these complexities, we conducted our experiments with due diligence, ensuring transparency and fairness in our reporting process.
>
> Furthermore, our aim was to encompass the majority of recently published methods by categorizing similar settings into different tables. This approach allowed us to compare our method with a larger array of methods, ensuring a fair and reliable comparison.
>
> Please kindly inform us if there is any issue with this viewpoint.
>
> ** *We've noted the revision in the rating and are keen to understand the specific factors that led to this adjustment. Your explicit feedback on the precise areas where you believe our responses fell short would be beneficial for us. We would like to address all concerns comprehensively and enhance the paper accordingly. If there are particular sections or aspects that you think still require attention or clarification, we would appreciate detailed insights.*

---

### Official Review · Reviewer_i2U9 · 2023-10-31

**Soundness:** 3 good
**Presentation:** 3 good
**Contribution:** 2 fair
**Rating:** 6
**Confidence:** 4

**Summary:**

This paper unveils the underlying factors of the trade-off between standard and robust generalization in adversarial training by examining the layer-wise learning capabilities of neural networks during the transition from a standard to an adversarial setting. The paper demonstrates that selectively updating specific layers while preserving others can substantially enhance the network's learning capacity empirically, and proposes a method to leverage a gradient prominence criterion to perform selective conservation, updating, and revision of weights named CURE. The paper verified the effectiveness of CURE on various dataset and architecture, which verifying the effect in enhancing the trade-off between robustness and generalization and alleviating robust overfitting empirically.

**Strengths:**

This paper has good originality, high quality and clear expression. The paper unveils the trade-off between standard and robust generalization in adversarial training in the perspective of layer-wise learning capabilities of neural networks and proposes a new method to alleviate robust overfitting.

**Weaknesses:**

The analysis of selective adversarial training are empirically not theoretically.It's better to provide theoretically analysis of selective adversarial training.

**Questions:**

1.Is the proposed method still works well on larger dataset, for example ImageNet?
2.There are so many hyperparameters such as α,β,γ,r,d, how to mediate so many hyperpapameters effectively?
3.For deeper neural networks, such as resnet-101, is the proposed method still works well?

---

> ### Author Response · Authors · 2023-11-19
> **Response to Reviewer i2U9 - 1/**
>
> We appreciate the reviewer's acknowledgment of the originality and high quality of our work and we would like to thank them for the valuable feedback. Our responses to the questions are below
>
> > The analysis of selective adversarial training are empirically not theoretically.It's better to provide theoretically analysis of selective adversarial training.
>
> We acknowledge the value of incorporating theoretical insights. In the current study, our focus was on addressing the challenges in adversarial training (AT) through behavioral analysis of layers. By delving into the learning dynamics during both standard and adversarial training via empirical analysis, we aimed to gain a clearer understanding of the underlying mechanisms. This empirical approach played a pivotal role in shaping the development of our proposed technique, aiming to improve the generalization trade-off and mitigate the overfitting phenomenon simultaneously.
> In this preliminary study, our emphasis has been on empirical observations. While our current work lays the foundation, we acknowledge the potential for theoretical studies to further enhance our understanding.
>
> > Larger dataset (ImageNet) and deeper neural networks (ResNet101)
>
> We appreciate the reviewer's feedback on additional results to show the efficacy of CURE. In the current literature, the prevailing datasets for adversarial training are CIFAR, SVHN, and MNIST, and the common architectures are ResNet18 and WideResNets (WRN34-10, WRN28-10). Unfortunately, among the 14 works we have compared with, none have demonstrated results on ImageNet or larger networks like ResNet101, also making it difficult  to obtain baselines for these.
>
> In response to the review, we have initiated additional experiments with larger networks for CURE and two other baselines. At this time, we present preliminary results for ResNet50-CIFAR10.  Due to the significant time and resource requirements, we will provide other results once these experiments are complete. We also add these additional results in Appendix (Table 6)
>
> |  | Nat | PGD20 | C&W | NRR |
> |---|---|---|---|---|
> | SAT | 72.86 | 37.54 | 36.92 | 49.00 |
> | TRADES | 73.06 | 40.33 | 37.56 | 49.61 |
> | CURE | 74.58 | 43.85 | 41.04 | 52.94 |
>
> > There are so many hyperparameters such as α,β,γ,r,d, how to mediate so many hyperpapameters effectively?
>
> The hyperparameters are stable across settings and datasets and also complement each other. Therefore, CURE does not require extensive fine-tuning across different datasets and settings. Some hyperparameters, like the revision rate (r) and decay factor (d) for the revision stage, are set universally (r = 0.2, d = 0.999) across all experiments,  The revision rate (r) determines the stochasticity of the revision stage, with a 20% probability of occurrence. This is more important during the initial stage of training and controlled reduction in revisions occurs as training progresses, as the network stabilizes. The hyperparameter γ, which is associated with the consistency regularization term, is generally stable and is set to 1.0.
>
> The main hyperparameters of interest are "p" and "α", which are quite intuitive. The parameter "α" influences the importance given to the adversarial distribution in the learning process. Higher values of "α" emphasize natural accuracy. Setting "α" to a lower value, such as 0.1 or 0.2, is preferable, especially when the network is pre-trained in the standard setting. Notably, "β" complements "α," where β = 1 - α (so we changed it in the equation 6)
>
> The "p" parameter is associated with estimating the percentile in gradients below which the gradients are set to zero. This percentile determines the fraction of weights that will not be updated, contributing more to the conservation of natural knowledge. The choice of "p" influences the balance between retaining previous knowledge and learning new information.
>
> We also add this in the Appendix (Table 10)
>
> | Sensitivity to gamma |  |  |  |  | Sensitivity to p |  |  |  |  | Sensitivity to alpha |  |  |  |  |
> |---|---|---|---|---|---|---|---|---|---|---|---|---|---|---|
> | p | alpha | gamma | Nat | Rob | p | alpha | gamma | Nat | Rob | p | alpha | gamma | Nat | Rob |
> | 30 | 0.1 | 1.0 | 86.76 | 54.92 | 30 | 0.1 | 1.0 | 86.76 | 54.92 | 30 | 0.1 | 1.0 | 86.76 | 54.92 |
> |  |  | 0.5 | 86.25 | 53.13 | 10 |  |  | 85.15 | 55.65 |  | 0.2 |  | 86.95 | 54.17 |
> |  |  |  |  |  |  |  |  |  |  |  | 0.5 |  | 88.52 | 53.67 |

---

> > ### Comment · Reviewer_i2U9 · 2023-11-22
> >
> > Thanks to the authors for their reply,I have no more questions.

---

> > > ### Author Response · Authors · 2023-11-22
> > > **Response to Reviewer i2U9**
> > >
> > > Thank you for your response. The experiments for ResNet101 have been completed, and we present the preliminary results below, as well as in Table 6 in the paper. It's important to note that these results are based on preliminary experiments, and we opted to use the ResNet18 hyper-parameters for training as we did not have a specific starting point from other works for these architectures.
> > >
> > > > ResNet101
> > >
> > > |  | Nat | PGD20 | C&W | NRR |
> > > |---|---|---|---|---|
> > > | SAT | 72.22 | 35.49 | 34.25 | 46.64|
> > > | TRADES | 72.28 | 39.16| 36.69| 48.67|
> > > | CURE | 73.67| 42.80 | 38.90 | 50.91|
> > >
> > > Please let us know us if there is any additional information we can provide to better highlight the effectiveness of CURE and also enhance your support for our work.. Thanks.

---

### Official Review · Reviewer_HvAu · 2023-10-31

**Soundness:** 3 good
**Presentation:** 4 excellent
**Contribution:** 4 excellent
**Rating:** 6
**Confidence:** 3

**Summary:**

This paper presents a novel approach to improving the adversarial robustness of DNNs while maintaining the performance on natural samples. The authors first conduct empirical studies to discover that updating weights in all layers in AT may be not good for the generalization of DNNs in both natural and adversarial samples. Thus, they propose an adaptive method to selectively update a subset of weights in the DNN. The proposed method is simple but empirically effective.

**Strengths:**

This paper presents a simple yet effective method of improving both robustness and generalization of DNNs.

This study discovers some interesting phenomena, e.g., updating middle layers is beneficial to standard and robustness generalization, and adversarial training increases the similarity between features of different layers.

The paper is well-written and easy to follow.

**Weaknesses:**

- The authors repeatedly use “overwritten” and “learning”, but I cannot grasp the essential difference between them. What kind of update of weights is referred to as “overwritten” and “learning”, respectively? Can you provide a clearer definition for the two terms? Besides, how do the authors come to the conclusion about “overwritten” and “learning” from the accuracy drop in Figure 2(a)?
- In Section 2, the authors update the weights of selected layers by re-initializing and updating them. Have you ever tried not re-initializing but directly finetuning them with adversarial examples? Learning weights from random noises, especially weights in shallow layers may lead to a significant drop in performance, but finetuning them will not.
- I’m a little confused about Figure 3. If the weights of deep layers (e.g. U-34) are updated while shallow layers (e.g. blocks 1 and 2) are fixed, the features in shallow layers are also supposed to be frozen. Thus, the similarity between shallow layers in U-34 should be the same as in the ST model. However, the similarities between shallow layers in Figure 3 are all different.
- Although this paper presents some differences in performance between updating different layers, it does not clearly “disentangle clean and adversarial representations” or “disentangle robust and non-robust features”. Also, this paper does not strictly define and extract layers with “greater learning capacity”. I suggest the authors use such claims carefully and seriously.
- Figure 4 indicates that updating high layers may cause robustness overfitting, but Figure 6 (b) shows a large ratio of updated gradients in high layers. I expect the authors to discuss more about such a misalignment.
- How about the representation similarity between different layers in the DNNs trained by CURE?

**Questions:**

- Is CURE used by training the model from scratch with the loss function in equation (5) or finetuning a pre-trained network? If the CURE is adopted on a pre-trained model, I am concerned that it may be unfair to compare it with other training-from-scratch methods. Besides, it also increases the computational cost to train the model twice (pre-train with ST and then finetune with CURE).
- How stable is CURE when using different hyper-parameters ($\alpha,\beta$ and $p$)?

---

> ### Author Response · Authors · 2023-11-19
> **Response to Reviewer HvAu - 1**
>
> We appreciate the reviewer's acknowledgment of the paper's strengths and thank them for their detailed insights and feedback.
>
> >  Can you provide a clearer definition for the two terms? “overwritten” and “learning”,
>
> In the context of our study, the term "overwritten" signifies a phenomenon where the knowledge acquired during standard training, primarily on natural images, undergoes a degree of compromise or replacement during adversarial training. This is akin to the observed behavior in continual learning, where adapting to a new task can lead to the erosion of previously acquired knowledge, prompting the exploration of dynamic and sparse architectures in that domain.
> Additionally, the inherent over-parameterization of deep neural networks implies that not all neurons operate at peak capacity, leaving room for learning new information from different distributions or data. Thus, we used these terms in the text.
>
> For empirical analysis, our objective was to analyze the behaviors of the network during the transition from standard training to adversarial training. The “Base” we start from is the network trained on natural images via standard training (ST). We use this network and then freeze different layers, re-initialize weights of the remaining and train on adversarial images. The observed decrease in test accuracy serves as one indicator of a potential overwriting of learned information.
> To enhance clarity, we have made adjustments to Fig 2(b), incorporating results for both standard training (ST) and standard adversarial training (SAT). This modification aims to visually represent the transition from ST to SAT by considering the perspective of layer re-initialization.
>
> > In Section 2, the authors update the weights of selected layers by re-initializing and updating them. Have you ever tried not re-initializing but directly finetuning them with adversarial examples?
>
> We acknowledge the suggestion, and we have provided the figure with results of directly fine-tuning selected layers with adversarial examples instead of re-initialization - Figure 11 in Appendix. Fine-tuning also results in a decline, as there is a sharp shift when the network encounters adversarial images after being trained only on natural images. While certain layers exhibit relatively higher performance than re-initialization, the overall trend remains the same.
>
> Further, our primary objective in the empirical analysis was to assess the behavior of the network through the lens of both re-initialization and updating. This approach allowed us to observe how the network adapts to adversarial training under different weight manipulation strategies, providing insights into the learning dynamics and the impact on performance.
>
> > Thus, the similarity between shallow layers in U-34 should be the same as in the ST model.
>
> Thanks for the keen observation. In Figure 3, we examine the similarity between the features generated when an adversarial sample is the input and the features when natural samples are the input for a specific network architecture. The figure illustrates the similarity patterns among different layers for nine models, each trained with distinct retention and update strategies. This pattern deviates from the standard (ST) model, as these networks receive adversarial samples as input, leading to changes in batch norm statistics (unlike in the ST model)
>
> There is also an observed "block structures" in these networks indicating a substantial level of intra-layer similarity and lower inter-layer similarity. For instance, in the U-1 model, the first block displays considerable dissimilarity with the fourth block, while the last three blocks exhibit higher similarity. In U-34, blocks 1 and 2 are fixed and hence have similarity between each other, while there are differences seen in the third and fourth block. Please let us know if more explanation is needed.
>
> > Paper does not disentangle "robust and non-robust features" and and extract layers with “greater learning capacity”.
>
> We acknowledge that the term "disentanglement" might introduce ambiguity. Figure 3 is not intended as a direct disentanglement method; rather, it serves as an illustrative representation. Our primary focus was not to disentangle features explicitly but to identify weights that can be updated to learn representations that are effective for both natural and adversarial images.
> Also, in reference to the term "layers with greater learning capacity," we meant CURE adopts a more automatic approach of choosing layers to be updated and not do it in the traditional engineering sense. We appreciate the reviewer's feedback and will make sure to remove these ambiguous terms in the revised manuscript for better clarity.

---

> > ### Author Response · Authors · 2023-11-19
> > **Response to Reviewer HvAu - 2**
> >
> > > misalignment between Figure 4 and figure 6(b)
> >
> > In our analysis, we observe various patterns that come into play. For generalization, as depicted in Figure 2, updating middle layers holds greater importance. When aiming for similarity between representations learned on standard and adversarial samples, later layers become more crucial. The observations from Figure 4 highlight the significant role of middle layers in balancing between reducing overfitting and maintaining higher test performance.
> >
> > Regarding Figure 6, he magnitude of updates remains higher in blocks 2 and 3 compared to block 4. The last high update is on batch-normalization layers rather than convolution layers, aligning with the dynamics of the decision boundary. Figure 10 in the Appendix specifically shows updates on conv layers, where the magnitude of updates is higher in middle layers.
> >
> > > the representation similarity between different layers in the DNNs trained by CURE?
> >
> > The representation similarity between different layers in the DNNs trained by CURE is visualized using the CKA metric in Figure 12. Compared to standard training (ST) and adversarial training (AT), CURE lies in the middle, leaning towards enhanced robust accuracy. It demonstrates a more balanced and nuanced performance, with an overall superior trade-off.
> >
> > > Is CURE used by training the model from scratch with the loss function in equation (5) or finetuning a pre-trained network?
> >
> > We understand the concern. Hence along with CURE, we had also introduced an efficient version of CURE, denoted as CURE{Eff} in Section A in Appendix. CURE{Eff} does not require a pre-trained model on natural images; instead, it undergoes a short warm-up phase where it is trained on natural samples before transitioning to adversarial training. Importantly, this entire training process, including the initial natural training and adversarial training, is completed within the same number of epochs (and hence has the same computational cost). The results presented in Table 4 in the Appendix showcase that CURE{Eff} performs nearly as well as CURE, with only a 1% lower performance.
> >
> > Additionally, for further comparison, we trained baselines such as Madry and TRADES using pre-trained models (trained in the standard training (ST) phase), and the results are available in Table5 in Appendix, and CURE still outperforms them.
> >
> > Our work serves as an initial exploration into understanding the diverse behaviors of neural networks and offers an effective solution. We believe that this method can inspire and drive future research towards developing more efficient designs.
> >
> > > How stable is CURE when using different hyper-parameters
> >
> > The hyperparameters are stable across settings and datasets and also complement each other. Therefore, CURE does not require extensive fine-tuning across different datasets and settings. Some hyperparameters, like the revision rate (r) and decay factor (d) for the revision stage, are set universally (r = 0.2, d = 0.999) across all experiments, The revision rate (r) determines the stochasticity of the revision stage, with a 20% probability of occurrence. This is more important during the initial stage of training and controlled reduction in revisions occurs as training progresses, as the network stabilizes. The hyperparameter γ, which is associated with the consistency regularization term, is generally stable and is set to 1.0.
> >
> > The main hyperparameters of interest are "p" and "α", which are quite intuitive. The parameter "α" influences the importance given to the adversarial distribution in the learning process. Higher values of "α" emphasize natural accuracy. Setting "α" to a lower value, such as 0.1 or 0.2, is preferable, especially when the network is pre-trained in the standard setting. Notably, "β" complements "α," where β = 1 - α (so we changed it in the equation 6)
> >
> > The "p" parameter is associated with estimating the percentile in gradients below which the gradients are set to zero. This percentile determines the fraction of weights that will not be updated, contributing more to the conservation of natural knowledge. The choice of "p" influences the balance between retaining previous knowledge and learning new information.
> >
> > We also add this in the Appendix (Table 10)
> > | Sensitivity to gamma |  |  |  |  | Sensitivity to p |  |  |  |  | Sensitivity to alpha |  |  |  |  |
> > |---|---|---|---|---|---|---|---|---|---|---|---|---|---|---|
> > | p | alpha | gamma | Nat | Rob | p | alpha | gamma | Nat | Rob | p | alpha | gamma | Nat | Rob |
> > | 30 | 0.1 | 1.0 | 86.76 | 54.92 | 30 | 0.1 | 1.0 | 86.76 | 54.92 | 30 | 0.1 | 1.0 | 86.76 | 54.92 |
> > |  |  | 0.5 | 86.25 | 53.13 | 10 |  |  | 85.15 | 55.65 |  | 0.2 |  | 86.95 | 54.17 |
> > |  |  |  |  |  |  |  |  |  |  |  | 0.5 |  | 88.52 | 53.67 |

---

> > > ### Comment · Reviewer_HvAu · 2023-11-22
> > >
> > > I thank the authors for their explanations. They have addressed most of my questions, but there are still some concerns.
> > >
> > > 1.  Can you provide a clearer definition for the two terms? “overwritten” and “learning”: I understand the meaning of "overwritten", but it is still unclear how to distinguish which weight changes represent the overwriting of features and which represent learning of features. In other words, when the accuracy drops in Figure 2, how did you identify which layers are being overwritten while other layers are learning new information? Or do you mean the drop in natural accuracy always reflects the overwriting of features? Given that these two terms frequently appear in the paper, I suggest the authors provide a clear and quantifiable definition for them.
> > >
> > > 2. misalignment between Figure 4 and Figure 6(b): I double-check Figure 6(b, bottom) and find that almost all high updates (peaks) happen in BN layers. In Figure 6(b, top), although conv layers are also updated, BN layers always have higher updated ratios. Could you further explain this phenomenon?

---

> > > > ### Author Response · Authors · 2023-11-22
> > > > **Response to Reviewer HvAu**
> > > >
> > > > > Terminology
> > > >
> > > > We acknowledge that the usage of the terminologies might be confusing. They are not distinct entities but rather describe the impact of weight changes on the accuracy of the distribution. The emphasis is on the fact that alterations in the weights of certain layers have a more pronounced effect on accuracy compared to others. For example, modifying layer 1 has a more significant impact on natural accuracy than modifying other layers. It results in a drop of accuracy of already learnt features, and this we refer to as overwriting.
> > > > However, when earlier layers are fixed, and only the weights of later layers are trainable, the drop in accuracy is less, as generalizable features from the early frozen layers still contribute to adapting to the new distribution. The impact of weight change is lesser in reference to reduction in performance, and we refer to it as having more learning capacity as the accuracy on adversarial improves and the natural does not drop too much.
> > > > The RGP metric provides a criteria based on both natural and adversarial objectives and helps determine the gradients that need to be fixed (or set to zero). Updating these weights will result in reduction of natural accuracy and hence should be fixed.
> > > >
> > > > To address potential confusion, we refine the phrasing of these terms in the introduction and also in empirical analysis in the paper. If further clarification is required or if there's any misunderstanding of the question on our part, please don't hesitate to inform us.
> > > >
> > > > >  In Figure 6(b, top), although conv layers are also updated, BN layers always have higher updated ratios. Could you further explain this phenomenon?
> > > >
> > > > Thank you for your insightful observation. The higher updates in BatchNorm layers compared to convolutional layers are indeed an intriguing result from our selective training analysis. The batchnorm statistics play a higher role in our case as we start from a natural model and train on the adversarial samples that have different data distribution. The higher updates in BatchNorm layers are consistent with findings in the literature examining the role of batch normalization in training. Notably, [1] reveals the potential for achieving remarkably high accuracy by training solely the affine parameters of BatchNorm and keeping all other parameters frozen at their original initializations. Moreover, [2] delves into various aspects of training in continual learning domains, suggesting that the impact of the batchnorm layer is data-dependent.
> > > >
> > > > We acknowledge the importance of this observation in our selective adversarial training analysis, and we are eager to delve deeper into understanding the specific role of BatchNorm in adversarial training. We also updated our paper in Section D.3
> > > >
> > > > [1] Training BatchNorm and Only BatchNorm: On the Expressive Power of Random Features in CNNs, Jonathan Frankle and David J. Schwab and Ari S. Morcos, ICLR 2021
> > > >
> > > > [2] Architecture Matters in Continual Learning, Seyed Iman Mirzadeh and Arslan Chaudhry and Dong Yin and Timothy Nguyen and Razvan Pascanu and Dilan Gorur and Mehrdad Farajtabar

---

> > > > > ### Author Response · Authors · 2023-11-23
> > > > > **Response to Reviewer HvAu**
> > > > >
> > > > > Thank you for all the insightful feedback. We would like to inquire if there are any remaining concerns regarding our work. Additionally, please let us know if there is any specific information we can provide to enhance your support for our research.

---

### Official Review · Reviewer_DhtN · 2023-11-01

**Soundness:** 2 fair
**Presentation:** 3 good
**Contribution:** 2 fair
**Rating:** 5
**Confidence:** 1

**Summary:**

This paper focuses on the trade-off between standard and robust generalization. To this end, this paper proposes CURE that leverages a gradient prominence criterion to perform selective conservation, updating, and revision of weights, which can tackle both memorization and overfitting issues.

**Strengths:**

The problem authors focused on is very interesting.

**Weaknesses:**

1. Experimental results in Fig. 2 cannot support authors' claim that ".. cause reduced performance on both data due to overwriting of learned information.." Specifically, just a comparison of accuracy in Fig. 2  cannot reflect the overwriting of learned information. Authors should design new solid experiments to support this conclusion.
2. Authors did not clarify how to disentangle robust and non-robust features, which still presents a significant challenge. Hence, Fig.3 is in doubt.
3. I wonder why "a subset with the most significant impact on accuracy" equals to the subset of weights that "contribute more to the joint distribution of both natural and adversarial accuracy." Can you prove it or explain it?
4. A algorithm flowchart will help readers better understand how weights are updated in each epoch. In each epoch, are different or same subsets of weights updated?
5. What does "sample∼U(0,1)<r" in Eq. 7 mean?
6. Experimental results cannot verify the effectiveness of the proposed CURE method, since authors just conducted experiments on resnet18 and resnet34. Please conduct more experiments on more classic DNNs.

**Questions:**

In this paper, many conclusions are not supported authors' experimental results, i.e., we cannot infer these conclusions just based on existing experimental results.
Thus, many conclusions in Section 3 are over-claimed.
Details are stated in weaknesses.

---

> ### Author Response · Authors · 2023-11-19
> **Response to Reviewer DhtN - 1/2**
>
> We would like to thank the reviewer for taking the time to review our work. Please find our responses below. We have also updated the paper and all the changes made in the paper are in blue.
>
> >Experimental results in Fig. 2
>
> We acknowledge the reviewer's concern regarding the interpretation of experimental results presented in Figure 2. Our objective was to analyze the behaviors of the network during the transition from standard training to adversarial training. The base, we start from is the network trained on natural images via standard training (ST). We use this network and then freeze different layers, re-initialize weights of the remaining and train on adversarial images. From Fig2, the network trained in ST has accuracy of ~95, and when only layer 1 is updated while keeping the rest of the network frozen, the natural accuracy drops to ~45. The observed decrease in test accuracy serves as one indicator of a potential overwriting of learned information. The analogy is also drawn with Continual Learning (CL) domain, where, as a network adapts to new tasks or distributions, there is a risk of overwriting previously acquired knowledge which is again measured only through the drop in accuracy. To further provide clarity, we also modify Fig 2(b) to include ST and SAT (which is the standard adversarial training) results, to show the transition from ST to SAT by examining the process of layer re-initialization.
>
> > disentangle robust and non-robust features
>
> We acknowledge that the term "disentanglement" might introduce ambiguity. Figure 3 is not presented as a direct disentanglement method; rather, it merely serves as an illustrative representation. Our primary objective was not to provide a definitive disentanglement mechanism but rather to showcase the transition as the network moves from standard to adversarial training. We intended to examine the similarity between the representations of natural and adversarial samples on differently trained networks. The network that has representations that exhibit similarity in both scenarios have the potential to perform effectively on both natural and adversarial images. This visualization helps emphasize the learning dynamics rather than providing a precise disentanglement methodology.
> In response to this feedback, we have removed such terms from the paper.
>
> > I wonder why "a subset with the most significant impact on accuracy" equals to the subset of weights that "contribute more to the joint distribution of both natural and adversarial accuracy
>
> The rationale behind our claim lies in the functionality of the Robust Gradient Prominence (RGP) metric. This metric is designed based on the premise that features with the most substantial impact on the model's predictions exert a greater influence on the gradients of the model parameters. The RGP score, as computed in Equation 6 of our paper, leverages gradients from both natural and adversarial examples. The coefficients α and (β = 1-α)  help balance the importance between natural and adversarial distributions. The RGP metric, by evaluating the gradients with respect to each weight, provides a quantitative measure of their importance in influencing both natural and adversarial accuracy.
> As we use the base model of ST to perform adversarial training, and view natural and adversarial images as distinct distributions, our objective is to update weights that can effectively learn from adversarial samples without compromising accuracy on natural images. The selected subset of weights, determined by the RGP metric, encapsulates those with the most significant impact on accuracy, facilitating a nuanced trade-off between natural and adversarial performance. This metric offers a quantitative measure of weight importance, guiding our approach to updating a subset of weights with a strategic focus on achieving optimal model performance across different data distributions.
>
> > A algorithm flowchart
>
> We appreciate the suggestion. Indeed, to enhance clarity, we have included a detailed algorithm in Appendix, Section C. In each epoch, the weights to be updated are chosen automatically by the RGP metric and can be different. CURE provides a dynamic mechanism to determine which weights to update and which ones to preserve in each layer of the network. There is no manual engineering of selecting the weights. As the training progresses, the RGP metric evolves and a more stable updation is seen during the final stages of training, as seen from in Fig6(b), where middle and later layers are updated more. This selective weight updating strategy is a key aspect of our method, allowing for targeted adaptation to adversarial data while preserving crucial information from natural data.

---

> > ### Author Response · Authors · 2023-11-19
> > **Response to Reviewer DhtN - 2/2**
> >
> > > What does "sample∼U(0,1)<r" in Eq. 7 mean?
> >
> > The revision model is stochastically updated by randomly sampling a scalar from a uniform distribution within the range of 0 to 1. This update occurs if the sampled scalar falls below the specified decay factor "r."
> > In our implementation, the value of "r" is set to 0.2. Consequently, the revision stage occurs stochastically with only a 20%  probability. Moreover, as training advances, we gradually decay the revision rate to minimize revisions when the network stabilizes.This revision stage is more crucial in the initial phases to prevent a drastic shift from natural to adversarial representations. We also added this in the algorithm in Appendix for more clarity.
> >
> > > more experiments on more classic DNNs.
> >
> > The prevalent architectures in AT literature are ResNet18 and WideResNets (WRN34-10, WRN28-10) and PreActReset18. We opted for these architectures as they are frequently featured in state-of-the-art papers, both historical and contemporary, providing comprehensive results and baselines. If there are specific architectures the reviewer would like to see, please feel free to specify, and we will consider them accordingly.
> >
> > Currently, we are introducing supplementary (preliminary) findings for CIFAR10-ResNet50, provided in the Appendix. As the baselines were not readily available for this architecture, we trained two baseline models, SAT (Madry) and TRADES, to establish a basis for comparison. We utilized the best hyperparameters determined for ResNet18 in all three methods respectively and applied them while training on ResNet50, ensuring a fair comparison within the given timeframe. The table is in Appendix (Table 6) and also presented below. Additionally, experiments for ResNet101 are currently in progress, and we will report the results accordingly.
> >
> > |  | Nat | PGD20 | C&W | NRR |
> > |---|---|---|---|---|
> > | SAT | 72.86 | 37.54 | 36.92 | 49.00 |
> > | TRADES | 73.06 | 40.33 | 37.56 | 49.61 |
> > | CURE | 74.58 | 43.85 | 41.04 | 52.94 |

---

> > > ### Author Response · Authors · 2023-11-23
> > > **Response to Reviewer DhtN - 3**
> > >
> > > We wanted to report that the ResNet101 experiments completed and hence we provide the results below and also in Table 6 in the paper. Kindly review the provided information, and please feel free to reach out if you have any further concerns
> > >
> > > > ResNet101
> > >
> > > |  | Nat | PGD20 | C&W | NRR |
> > > |---|---|---|---|---|
> > > | SAT | 72.22 | 35.49 | 34.25 | 46.64|
> > > | TRADES | 72.28 | 39.16| 36.69| 48.67|
> > > | CURE | 73.67| 42.80 | 38.90 | 50.91|

---

### Author Response · Authors · 2023-11-19
**General Response**

We would like to thank all the reviewers for dedicating their time and effort to assess our work and share insightful feedback. We've made the recommended changes and submitted the revised paper, *highlighting modifications in blue for clarity.*

**Goal** : To re-iterate, our goal was to enhance the trade-off between standard and robust generalization. We drew valuable insights from our empirical study that analyzes learning behavior of  neural networks during standard and adversarial training, and designed a selective adversarial training algorithm CURE. CURE facilitates effective learning from both natural and adversarial data distributions, enhancing the trade-off. Our method's efficacy is demonstrated across various settings, including the mitigation of the robust overfitting issue. Baseline results are obtained from original works, and our experiments prioritize transparency and fairness.

**Changes**:

- Figure 2b - modified to include standard and adversarial training accuracy for better clarity
- Algorithm - Section C in appendix

` Additional Results:`
- Table 5 - Effect of Pre-training on other Baselines
- Table 6  - Results on ResNet50 and ResNet101 architecture
- Table 7 - Ablation Study
- Table 8 - Comparison with NRR calculated using Natural and AutoAttack
- Figure 9 - Robustness against all natural corruptions (without grouping)
- Table 10 - Hyperparameter Sensitivity
- Figure 11 - The same analysis as Figure 2 performed without re-initializing layers (but with fine-tuning)
- Figure 12 - Similarity analysis on CURE-trained network

Apart from these results, we have made modifications in text based on the minor comments and feedback.
Please let us know if  if further clarification is required or if there are specific aspects you would like more information on. Thanks.

---

### Meta-Review · Area_Chair_7FAU · 2023-12-06

**Metareview:**

This paper introduces CURE, a new method to improve the trade-off between standard accuracy and robust accuracy when adversarially fine-tuning a pre-trained model. The paper first analyzes the impact of updating different layers of the model while keeping others fixed.  Based on the observations, the paper proposes a method that leverages a gradient prominence criterion to dynamically perform selective conservation, updating, and revision of model weights. The efficacy of this method is validated through experiments across various datasets and model architectures.
Reviews (HvAu, i2U9, PH92) agree that the identified phenomenon for layer-wise learning is insightful, and the proposed learning method is interesting. During rebuttal, reviewers DhtN and HvAu pointed out the ambiguous terminology used in the paper and the unsupported claims from the figure. The authors clarified the questions and updated the corresponding text and figures in the paper. Review HvAu acknowledged that most of the questions have been resolved. Additional concerns regarding the method's generalizability to larger datasets and models were brought up by reviewers DhtN and i2U9, to which the authors responded with additional results confirming the method's effectiveness. Furthermore, reviewer PH92 raised concerns about unfair evaluation settings, including the attack method being used, the miss of evaluation metrics used in previous papers, and the inconsistent loss function between baselines and the proposed method. The authors provided additional results that include the suggested attack method and evaluation metric, and also clarified that the comparison with baselines is consistent. Finally, reviewer PH92 also expressed concerns about the distinctions with existing works, due to the use of an existing objective function. The authors explained that the uniqueness of the paper is the proposed selective training approach, and the common objective function was also used in many previous works. The AC noted these explanations and felt they reasonably addressed the concerns.
Given the overall positive reception of the presented phenomenon and proposed method, we recommend acceptance of the paper. However, if accepted, the authors need to address the negative reviews in the revision, add suggested evaluations, and improve the clarity of the writing.

**Justification For Why Not Higher Score:**

The paper did not receive a higher score due to concerns raised by the reviewers, including ambiguous terminology, unsupported claims in figures, questions about the method's generalizability, and potential issues with unfair evaluation settings. While the authors addressed most of these concerns during the rebuttal phase, other questions persist such as the inconsistent comparison with baselines.

**Justification For Why Not Lower Score:**

The paper was not given a lower score because of the overall positive reception from reviewers regarding the insightful layer-wise learning phenomenon it identified and the interesting proposed learning method. The authors effectively addressed key issues raised during the rebuttal phase, by updating the text and figures in the paper. The authors also responded proactively to additional concerns about the model’s generalizability to larger datasets and models and evaluation settings, by providing additional results and clarifications.

---

### Decision · Program_Chairs · 2024-01-16

Accept (poster)